# Revealing Ancient Wheat Phylogenetic Diversity: Machine Learning and Logistic Regression Identify *Triticum sphaerococcum* in Bronze Age Iberia

**DOI:** 10.3390/genes16121477

**Published:** 2025-12-09

**Authors:** Diego Rivera, Milagros Ros-Sala, Diego-José Rivera-Obón, Francisco Alcaraz, P. Pablo Ferrer-Gallego, Emilio Laguna, Nikolay P. Goncharov, Yulia V. Kruchinina, Concepción Obón

**Affiliations:** 1Departamento Biología Vegetal, Facultad Biología, Universidad de Murcia, 30100 Murcia, Spain; falcaraz@um.es; 2Department of Prehistory, Archaeology and Ancient History, University of Murcia, 30110 Murcia, Spain; milaros@um.es; 3RITM (Research Center in Economics & Management), Université Paris-Saclay, 54, Boulevard Desgranges, 92330 Sceaux, France; diego-jose.rivera-obon@universite-paris-saclay.fr; 4Servicio de Vida Silvestre y Red Natura 2000, Centro para la Investigación y Experimentación Forestal (CIEF), Generalitat Valenciana, Avda. Comarques del País Valencià 114, Quart de Poblet, 46930 Valencia, Spain; flora.cief@gva.es; 5Independent Researcher, c/Moncada, 26, Catarroja, 46470 Valencia, Spain; emilio.laguna@outlook.es; 6Institute of Cytology and Genetics, Siberian Branch of Russian Academy of Sciences, Novosibirsk 630090, Russia; gonch@bionet.nsc.ru (N.P.G.); kruchinina2023@yandex.ru (Y.V.K.); 7Instituto de Investigación e Innovación Agroalimentario y Agroambiental (CIAGRO), Escuela Politécnica Superior de Orihuela, Universidad Miguel Hernández, Ctra. Beniel, Km 3,2, Orihuela, 03312 Alicante, Spain; cobon@umh.es

**Keywords:** archeological wheat taxonomy, machine learning in archaeobotany, *Triticum* grain classification, morphometric analysis, Argaric Early Bronze Age, Random Forest classification, logistic regression, Mediterranean prehistoric agriculture, computational archaeobotany, grain dimensional analysis

## Abstract

Background/Objectives: Identifying archaeobotanical wheat remains is central to reconstructing the evolutionary history of cereal crops. Beyond documenting agricultural practices, such analyses provide critical evidence of phylogenetic diversity, lineage persistence, and local extinction events within the genus *Triticum* L. This study applies advanced computational morphometrics to reveal deep-time changes in wheat species distribution, including the disappearance of taxa now phylogeographically confined to central Asia. Methods: We developed a machine learning framework integrating Random Forest compared with logistic regression to classify morphometric data from 848 dry and 340 experimentally carbonized modern grains representing multiple wheat taxa (genus *Triticum*), alongside 15 archaeobotanical *T. turgidum* subsp. *parvicoccum* and 38 *T. aestivum* var. *antiquorum.* This probabilistic classifier was then applied to 2463 archeological wheat grains, including 48 from Punta de los Gavilanes and 517 from Almizaraque (southeastern Spain, 3rd–2nd millennium BC). Results: The analysis identified *Triticum sphaerococcum* and other phylogenetically distinct wheat taxa—today restricted to central and south Asia—among western European Bronze Age assemblages. These findings indicate that lineages now regionally extinct once formed part of a broader cultivated gene pool spanning into the western Mediterranean. Morphometric evidence highlights that past wheat diversity encompassed multiple clades and morphotypes absent from modern European germplasm. Conclusions: Our results demonstrate substantial phylogenetic turnover in wheat over the past 4000 years, marked by regional extirpations and contraction of once-widespread lineages to central Asia. This provides rare archeological evidence for the tempo and mode of cereal phylogeography, illustrating how domesticated lineages underwent extinction and range restriction akin to wild taxa. By integrating computational morphometrics with archaeobotanical evidence, this study establishes a scalable framework for tracing cryptic phylogenetic diversity, refining models of wheat domestication and assessing long-term genetic erosion in cultivated plants.

## 1. Introduction

### 1.1. Research Context and Significance

Archaeobotany examines plant remains from archeological contexts to elucidate past human–plant interactions, subsistence economies, and environmental conditions [1]. Among these remains, seeds and fruits are particularly diagnostic, offering key evidence for dietary practices, agricultural development, trade networks, and ecological shifts [2]. Their identification relies on morphological characteristics—such as size, shape, and surface texture—enabling taxonomic classification to species or genus level, which is critical for reconstructing past societies with precision.

Seeds and fruits are frequently preserved through carbonization, waterlogging, or desiccation, making them invaluable for tracing crop domestication and dispersal [3]. For instance, the spread of staple crops like wheat, maize, and rice has been mapped through archaeobotanical analysis [4]. Additionally, wild plant remains reveal foraging strategies and land-use patterns, while exotic species indicate long-distance exchange [5]. Ecological inferences can also be drawn from habitat preferences of identified taxa, contributing to paleoenvironmental reconstructions [6].

However, methodological challenges complicate identification. Preservation biases, such as carbonization-induced distortion or waterlogged fragmentation, obscure diagnostic features [7]. Taxonomic overlap among morphologically similar species necessitates advanced techniques, including scanning electron microscopy (SEM) and geometric morphometrics [8]. Furthermore, reliance on modern reference collections may overlook ancient morphological variation and extinct taxa, limiting accuracy [9]. Taphonomic processes, including bioturbation and burning, further degrade remains, necessitating careful contextual interpretation [10].

To address these challenges, archaeobotany integrates traditional and innovative approaches. Comparative morphology, supported by digital databases like the *Digital Seed Atlas of the Netherlands* and *The Digital Plant Atlas* [11,12], remains foundational. High-resolution microscopy and SEM enhance surface-detail analysis, while computational methods, such as machine learning, improve classification [13]. Interdisciplinary collaboration—incorporating ancient DNA, stable isotope analysis, and ethnobotany—strengthens interpretive frameworks [14].

In summary, seed and fruit identification is central to archaeobotanical research, bridging past human behavior and environmental change. Despite inherent difficulties, methodological advancements and interdisciplinary synergies continue to refine analyses, ensuring these remains remain pivotal in understanding ancient subsistence, ecology, and cultural evolution [15].

### 1.2. Archeological and Botanical Background

Since the early 19th century, archaeobotanical research has prioritized the identification of plant remains, particularly cultivated seeds and fruits, yielding thousands of publications [16,17,18,19]. The identification of archaeobotanical materials, notably seeds and fruits, is usually based on the comparison with dry or experimentally carbonized modern materials. However, other possibility is the existence of archaeobotanically based taxa presently extinct. While most studies classify specimens using extant taxa, some archaeobotanical material has warranted new taxonomic designations, exemplified by *Triticum parvicoccum* Kislev [20,21,22].

Archaeobotany occupies a unique interdisciplinary niche between paleontology, botany, and agronomy, necessitating rigorous adherence to the *International Code of Nomenclature* (Madrid Code) to avoid ambiguities in fossil/non-fossil classifications [23]. Holocene specimens are typically treated as recent (non-fossil) material, though palaeobotanical contexts may term “subfossils” taxonomically subordinate to living taxa [24].

### 1.3. Challenges in Prehistoric Grain Identification

A persistent challenge arises with small-grained naked wheat from archeological sites, which long lacked modern analogs, prompting extinct-type taxa proposals [25,26]. The genus *Triticum* L. (Poaceae: Triticeae Dumort.) comprises ~25 wild and domesticated species [27], yet its classification remains contentious despite molecular tools, due to its recent domestication (6000–12,000 BP) [28,29]. Current taxonomy divides the genus by ploidy levels (di- (2*n* = 2*x* = 14), tetra- (2*n* = 4*x* = 28), hexaploid (2*n* = 6*x* = 42)), cytoplasm, and genome structure [30,31,32,33,34,35], with sections *Monococcon* Dumort., *Dicoccoides* Flaksb., *Triticum, Timopheevii* A.Filat. et Dorof., *and Compositum* N.P.Gonch., genetically validated [33,34]. However, some argue for segregating ploidy levels into distinct genera [36].

Kislev (1979/1980) initially described *T. parvicoccum* Kislev as a new tetraploid naked wheat species based on subfossil material from Near Eastern archeological sites, characterized by small grains and short, narrow internodes of rachis (spike axis) [37]. Subsequent studies contested this classification, suggesting it should be relegated to a subspecies of *T. turgidum* [35,38,39,40,41,42]. Notably, Kislev (2009) later treated it as *T. turgidum* subsp. *parvicoccum* without formally validating the name [43], a resolution finally addressed in Rivera et al. (2025) [44].

Schultze-Motel documented the spatiotemporal distribution of *T. parvicoccum*, with findings spanning west Asia, the Balkans, and Transcaucasia from the 7th millennium BCE to the 12th century CE [21]. This synthesis omitted detailed records from the western Mediterranean, leaving its presence in this region unconfirmed [45].

Direct morphological comparisons between archeological and modern plant specimens require caution due to temporal discontinuities and preservation biases. This limitation is exemplified by *T. vulgare* var. *antiquorum* Heer [46], originally described from Swiss Neolithic grain remains [25,26]. Although similar living specimens were collected in Mountainous Tajikistan (1969, 1982), their taxonomic equivalence to the ancient material remains uncertain [46,47]. Udachin [47] later elevated this taxon to species rank as *T. antiquorum* (Heer) Udachin.

Flaksberger in 1930 [48] reclassified this wheat as *T. compactum* var. *antiquorum*, noting its hexaploid characteristics and dense spikes, while distinguishing it from typical *T. compactum* by its spherical grains.

Finally, Goncharov in Rivera et al. [49] combined the taxon from central Asia as *T. sphaerococcum* subsp. *antiquorum* N.P.Gonch. and the archaeobotanical taxon (*T. aestivum* (subsp. *compactum***)** var. *antiquorum* (Heer) H. Messik.) was lectotypified.

The identification of small-grained wheat from Early Bronze Age Argaric sites presents unique challenges due to the limited morphological comparators available. Current diagnostic approaches must rely on modern wheat taxa, which may not fully capture ancient phenotypic variability due to millennia of domestication and breeding [2], and two archaeobotanically defined taxa (extinct), preserved only in the archeological record [37,38,39,40,41,42,43].

This constrained taxonomic framework reflects the evolutionary discontinuity between ancient and modern wheat populations and the potential extinction of early cultivated forms during agricultural intensification and the methodological necessity of creating archaeo-taxa for specimens without clear modern analogs [7].

The situation parallels challenges seen in other Mediterranean Bronze Age contexts, where “orphan” wheat morphotypes require archaeobotanical rather than neobotanical classification systems [24,25,26,45,49]. This approach acknowledges both the distinctiveness of ancient crop species and the limitations of direct morphological comparison across temporal divides.

### 1.4. The Gavilanes Site

The southeastern Iberian Peninsula features numerous Bronze Age settlements, including the coastal site of Punta de los Gavilanes (Mazarrón Bay, Murcia, Spain) (Figure 1) where small-grained wheat remains, tentatively identified as ‘*T. parvicoccum’* and ‘*T. antiquorum*’, were recovered. This strategic promontory (6.30 m.a.s.l.), composed of dolomites and limestones, existed in a saline coastal environment (Figure 2). Occupied from the late 3rd millennium (22nd century cal. BC, 3855 ± 30 BP) through Roman times (Appendix A), the site shows continuous habitation by Argaric Early/Late Bronze Age populations [50]. Phase IVa-c (3855 ± 30 to 3360 ± 35 BP), associated with Argaric Bronze culture, demonstrates an economy combining marine resource exploitation with agriculture, including plant processing for consumption [50,51,52,53].

Initial occupation coincided with warmer, more humid climatic conditions that enhanced local biodiversity, persisting through Phase IV’s early horizons [53]. Stratigraphic analysis reveals four chrono-cultural phases, with Phase IV (Early Argaric Bronze Age) being the oldest and the focus of this study [50,51,52]. The carpological assemblage includes both cultivated and wild species, notably primitive wheat and barley varieties, providing novel insights into 3rd–2nd millennium BC (2200–1500 cal. BC) agriculture [54]. The collection includes specimens of eight species of shrubs and trees, notably featuring *Coronilla talaverae* Lahora and Sánchez-Gómez, a winter-flowering shrub that is currently listed as endangered and has been extirpated from the Punta de los Gavilanes area. Seeds of this endemic *Coronilla* species have been found in association with materials dated to approximately 3900 cal. years BP, within a Bronze Age cultural context. The local extinction of this species was likely due to anthropogenic alterations of its natural habitat or environmental changes [54].

The recovery of wheat remains at the Punta de los Gavilanes site is strongly associated with key domestic and productive contexts, providing insights into the role of cereals within the settlement’s subsistence strategies. As illustrated in Figure 3, the principal contexts where wheat remains were identified include roasting structures, domestic drying/smoking facilities, and habitation spaces.

The roasting structure located within a 1TM house (Figure 3A) suggests that cereals may have been processed through thermal treatment, for consumption or preservation. Similarly, the presence of a domestic drying/smoking facility within a 1TS house (Figure 3C) indicates that controlled exposure to heat was employed, potentially to extend the shelf life of stored grains or facilitate further processing. Additionally, an esparto basket recovered in a 2TM building (Figure 3B) represents a key storage element, used for keeping cereals or other agricultural products. The spatial distribution of these features within the settlement’s urban layout, as depicted in Figure 3D, highlights the integration of food processing and storage areas within domestic units across the upper and middle terraces.

These findings underscore the importance of wheat at Punta de los Gavilanes, not only as a dietary staple but also as a component of broader food preparation and storage practices. The association of cereal remains with these specific contexts suggests a well-organized subsistence economy in which thermal processing and storage played a critical role.

Archaeobotanical remains from Bronze Age Argaric sites (2200–1500 cal. BC), notably Punta de los Gavilanes (Mazarrón, Spain), in southeastern Iberia present an opportunity to investigate ancient wheat diversity and test hypotheses about early agricultural genetics in this peripheral European region. Based on preliminary morphological observations suggesting the potential presence of *T. sphaerococcum* as a predominant species, alongside other wheat taxa including T. *aestivum* and rare archaeobotanical forms, this study aims to

Evaluate the hypothesis that *T. sphaerococcum* was present in Bronze Age Iberian agriculture through comprehensive morphometric analysis, comparing archeological specimens with modern reference material using established diagnostic criteria.

Identify and characterize the wheat assemblage composition to test whether the observed morphological diversity represents distinct taxa, potentially including *T. aestivum*, *T. aestivum* subsp. *compactum* var. *antiquorum*, and *T. turgidum* subsp. *parvicoccum*, within Argaric agricultural systems.

Evaluate the biogeographical hypothesis that diverse wheat taxa could have been successfully cultivated in coastal southeastern Iberia during the Bronze Age, examining their potential adaptive significance under contemporary climatic conditions and implications for understanding wheat dispersal routes.

Assess the agricultural and evolutionary implications of the identified wheat assemblage for regional crop diversification strategies, testing whether the presence of multiple taxa supports hypotheses about maritime exchange networks facilitating wheat genetic diversity in peripheral European agricultural zones.

These hypotheses, if supported, would contribute significant new data to our understanding of ancient crop genetic diversity and provide insights into the evolutionary history of wheat cultivation in Mediterranean Europe, with implications for both archeological genetics and crop conservation efforts.

## 2. Materials and Methods

### 2.1. Sample Collection and Identification and Datation Techniques for Punta de los Gavilanes

Seventy-six archaeobotanical samples were systematically recovered from various excavation phases conducted between 2004 and 2010, utilizing standard flotation techniques. These samples originated from diverse functional contexts including roasting ovens, milling areas, domestic hearths, kitchen structures, an esparto basket, and various floor sediments (Figure 3).

The samples underwent processing at the Laboratory of Ethnobotany of Murcia University through a calibrated sieve column with mesh diameters ranging from 2 mm to 0.1 mm. Subsequent laboratory analysis employed several optical instruments: an Olympus SZ-11 trinocular stereo microscope mounted on an SZ-STU 1 articulated arm with an SZ-PT adapter and Leica EC3 camera. Digital imagery was captured using an Arcus 1200 Agfa scanner, while scanning electron microscopy was performed with a JEOL JSM-6.100 instrument.

For taxonomic identification, recovered fruit and seed remains were compared against the laboratory’s carpological reference collection. The primary identification framework assumed that species documented in the modern Flora of Murcia from the late 20th and early 21st centuries [55,56,57,58] would encompass the archeological taxa recovered. However, the distinctive characteristics of small-grained wheat specimens necessitated specific comparison with archaeobotanical and modern taxa. Taxonomic nomenclature was standardized according to multiple authoritative databases [59,60,61,62].

Seed dimensions analysis was conducted using a Mitutoyo caliper. Comparative dimensional data for “*T. parvicoccum*” were derived from multiple sources [21,37,41,59,63], encompassing reference materials from west and central Asia and the Mediterranean region and those of “*T. antiquorum*” were derived notably from Flaksberger [48] and Heer [26] (Table 1). Diagnostic morphological characters and comparative analysis with other archeological wheat samples relied on established criteria from Goncharov and Gaidalenok [64], Hillman et al. [65], Hillman [66], and Jacomet [39].

Radiocarbon determinations were calibrated using CLIB 7.1 [67].

**Table 1 genes-16-01477-t001:** Wheat caryopses used as training set (dry modern, carbonized modern, and archaeobotanical type material) and archeological material analyzed.

Taxa	NT	Type	Provenance	MV	RF	LO
**Desiccated modern caryopses**						
*Triticum aestivum* L. subsp. *aestivum, T. aestivum* subsp. *compactum* (Host) H.Messik., *T. aestivum* subsp. *macha* (Dekapr. and Menabde) McKey*, T. aestivum* subsp. *spelta* (L.) Thell.*, T. monococcum* L. subsp. *monococcum, T. monococcum* subsp. *aegilopoides* (Link) Thell., *T. sphaerococcum* Percival subsp. *sphaerococcum *(*)*, T. sphaerococcum* subsp. *antiquorum* N.P.Gonch., *T. turgidum* L. subsp. *turgidum, T. turgidum* subsp. *carthlicum* (Nevski) Á.Löve and D.Löve*, T. turgidum* subsp. *dicoccoides* (Asch. and Graebn.) Thell., *T. turgidum* subsp. *dicoccum* (Schrank ex Schübl.) Thell., *T. turgidum* subsp. *durum* (Desf.) Husn., *T. turgidum* subsp*. georgicum* (Dekapr. & Menabde) Mac Key ex Hanelt*, T. turgidum* subsp. *polonicum* (L.) Thell., *T. turgidum* subsp. *turanicum* (Jakubz.) Á.Löve & D.Löve	848	TS	Tellez and Ciferri [68] and Institute of Cytology and Genetics, Siberian Branch of Russian Academy of Sciences, Novosibirsk	0	848	848
**Carbonized modern caryopses**						
*T. aestivum* L. subsp. *aestivum, T. aestivum* subsp. *compactum* (Host) H.Messik*., T. aestivum* subsp. *spelta* (L.) Thell.*, T. monococcum* L. subsp*. monococcum, T. sphaerococcum* Percival subsp. *sphaerococcum, T. turgidum* L. subsp*. turgidum, T. turgidum* subsp. *dicoccum* (Schrank ex Schübl.) Thell.	340	TS	Tellez and Ciferri [68]	0	340	340
**Archaeobotanical types**						
*T. aestivum* var. *antiquorum* (Heer) H.Messik.	38	TS	Flaksberger [48] and Heer [25,26]	3	38	38
*T. turgidum* subsp. *parvicoccum* Kislev (Type)	3	TS	Tell Batash [41]	3	3	3
*T. turgidum* subsp. *parvicoccum* Kislev	15	TS	Deir Alla, Tell Keisan [21,37,59,63]	2	15	15
**Archeological caryopses from Punta de Gavilanes**						
	48	OT	Punta de los Gavilanes	48	48	48
**Archeological caryopses from Almizaraque**						
	517	OT	Tellez and Ciferri [68]	0	517	517
**Other archeological caryopses**						
	2463	OT	[25,26,37,48,68,69,70,71,72,73,74,75,76,77,78,79,80,81]	31	0	0

Abbreviations: N, total number of caryopses included in the study; TS, training set; OT, other archaeobotanical material; MV, caryopses used in the multivariate analyses; RF, caryopses used in the Random Forest; LO, caryopses used in the logistic regression. Note: * While recognizing that the most widely used and accepted combination for scientifically naming Indian Dwarf Wheat is *T. aestivum* subsp. *sphaerococcum* (Percival) Mac Key and considering the recent combination for a related wheat subspecies from Tajikistan as *T. sphaerococcum* subsp. *antiquorum* N.P.Gonch., in order to maintain nomenclatural coherence, we use, for the type of *T. sphaerococcum* Percival, the combination *T. sphaerococcum* subsp. *sphaerococcum*.

### 2.2. Weat Caryopses Used in the Analyses

A total of over 4000 wheat caryopses were included in the analysis, comprising modern desiccated reference specimens, modern experimentally carbonized material, archaeobotanical type specimens, and other archaeobotanical caryopses, in addition to assemblages from the Punta de los Gavilanes and Almizaraque archeological sites (Table 1).

Reference collections of data from modern materials desiccated (dry) or experimentally carbonized were obtained notably from the work of Tellez and Ciferri [68], who also supplied numerous samples from Spanish archeological sites. Tellez and Ciferri [68] sourced modern wheat specimens from the living collection of the Centro de Cerealicultura at the Instituto Nacional de Investigaciones Agronómicas. They conducted experimental carbonization by subjecting grain samples to a temperature of 200 °C for 24 h in a controlled oven environment. The carbonization process induced differential morphological alterations across wheat species; notably, *T. sphaerococcum* exhibited no significant variation in length, while *T. turgidum* and *T. aestivum* maintained consistent depth measurements. The carbonization protocol resulted in increased sphericity of the grain morphology [68].

### 2.3. Multivariate Cluster Analysis

This study examined a sample of 48 archeological grain specimens from Punta de los Gavilanes, compared with 3 *T. aestivum* var. *antiquorum*, 3 *T. turgidum* subsp. *parvicoccum* from Tell Batash and one each from Tell Keisan and Deir Alla, and 31 specimens from other archeological sites (Table 1). The analysis focused on the following morphometric variables: Length, Breadth, Depth, and the derived allometric ratios Length/Breadth, Length/Depth, Breadth/Depth, and 100×(Breadth/Length).

To quantify inter-sample dissimilarity, the Mean Euclidean dissimilarity index, as described by Perrier et al. [82] and Perrier and Jacquemoud-Collet [83], was employed. This index evaluates the contribution of each variable (*x_ik_)* to the total sum across all variables (*x_i_*), thus facilitating a comparative analysis of specimen profiles. The dissimilarity between units *i* and *j* (*d_ij_*) is calculated as (1):
(1)$$d_{ij}=\sqrt{\frac{1}{K}\sum_{k=1}^{K}{\left(x_{ik}-x_{jk}\right)}^{2}}\;\text{for}\;j\;\neq \;i.$$
where *i*, *j* = 1, 2, ..., *N* (samples, rows), *N* = 87; and *k* = 1, 2, ..., *K* (variables, columns). A *d_ij_* value of 1 indicates complete dissimilarity between samples *i* and *j* across all variables, while a value of 0 signifies identity.

The resulting pairwise dissimilarity matrix was then subjected to multidimensional analysis. To visualize the relationships among samples in a two-dimensional space, cluster analysis was performed. Specifically, Ward’s minimum variance method, which aims to minimize within-cluster variance, was utilized to generate a hierarchical clustering solution [84]. Additionally, Weighted Neighbor Joining analysis with 500 bootstrap replicates was conducted to assess the robustness of the identified clusters.

The application of distance-based phylogenetic trees for the taxonomic assignment of archeological seed samples is well established. For instance, Pagnoux et al. [85] successfully assigned archeological grape seeds using UPGMA clustering based on Mahalanobis distances. Similarly, Rivera et al. [86] employed Ward’s clustering to tentatively classify archeological *Phoenix* seed samples.

The resulting dendrograms were visualized using FigTree software version 1.4.4 [87].

### 2.4. Machine Learning Methodology

For this research we used in the training set 848 modern desiccated caryopses, 340 modern carbonized, and 56 archaeobotanical; details of the taxa can be found in Table 1. We identified 48 caryopses from Punta de los Gavilanes and 517 from Almizaraque.

In this study, we employed logistic regression models [88] and the Random Forest technique [89,90,91,92] to assess the probability that individual seeds belong to a specific taxon based on morphometric data and comparison collections. Additionally, we investigated cases where seeds exhibit intermediate characteristics that hinder clear taxonomic classification.

The analysis was conducted using R Studio (c2025.05.1+513.pro3) within the R statistical computing framework [93]. R is an open-source software environment for statistical analysis and visualization, while RStudio [94] provides an integrated development environment (IDE) designed to enhance productivity in both R (13 4.4.0) and Python (3.13). The tools used are available through the CRAN R Project [95].

Logistic regression links the conditional probability of a binary response variable to explanatory variables [88,96]. In this case, we estimated the probability of a seed being classified as a determined taxa based on dimensions and allometric indices. In R, the logistic regression model (Logit) is implemented through the generalized linear model function. Logit models are a subset of generalized linear models in which the response variable follows a binomial distribution, and the link function is the logit function, which represents the logarithm of the odds ratio. In our study, the Logit model in R assigned a probability value (LOG) ranging from 0 to 1 to indicate the likelihood of each seed belonging to one of the available taxa in the reference collection.

While traditional machine learning algorithms often suffer from low classification accuracy and overfitting, Random Forest overcomes these limitations. It is an ensemble learning method that aggregates multiple decision trees, with each tree contributing a unit vote toward the final classification outcome. This approach enhances classification accuracy, is robust to noise and outliers, and mitigates overfitting [92]. Random Forest has been widely adopted in data mining and biological research.

For this analysis, we utilized the R package ‘random Forest’ version 4.7-1.1 [90,91,92], which implements the Random Forest algorithm as proposed by Breiman in 2001 [89]. This technique constructs a forest of decision trees using random input selections. A key advantage of Random Forest is its ability to manage a large number of variables while identifying the most significant attributes. The algorithm employs parameters such as *ntree* (default: 500 trees) and *mtry*, which determines the number of variables randomly selected at each split. Since our study was regression-based, weight was set to null and *replace = true* was applied to prevent overfitting [92]. The model generated probability values (RF) ranging from 0 to 1, indicating the likelihood of allocation to each taxon for each seed. Furthermore, we explored the application of Random Forest to evaluate the relative influence of variables in distinguishing taxa.

As noted by Couronné et al. in 2018 [96], the performance of logistic regression and Random Forest classifications is significantly influenced by the inclusion criteria used for dataset selection. Therefore, it is crucial to explicitly define dataset selection processes. In this study, we classified all dry and experimentally carbonized example seeds based on their sample origins as follows:

Training datasets:•Modern dry: Unaltered samples from the reference collection, attributed to different *Triticum* taxa.•Modern carbonized: Experimentally carbonized samples, initially dry, attributed to different *Triticum* taxa in the reference collection.•Archaeobotanical taxa: Includes *T. turgidum* subsp*. parvicoccum* and *T. aestivum* var. *antiquorum.*

Excluded categories: Further archeological samples were not included in the training datasets.

This approach ensured a well-defined classification process while mitigating biases introduced by dataset selection criteria.

## 3. Results

### 3.1. Taxonomic Classification Outcomes

The phenetic analysis of archeological wheat grains from Punta de los Gavilanes, utilizing Weighted Neighbor Joining (WNJ), with 500 bootstraps, and Ward’s clustering methods (Figure 4), reveals distinct genetic groupings indicative of divergent domestication pathways. The study also identifies two primary archeological taxa: *T. antiquorum* and *T. parvicoccum*, represented by type material from Swiss Neolithic sites and Tell Batash, respectively (included in the training set, Table 1).

*T. antiquorum* exhibits significant diversification associated with Gavilanes samples, evidenced by multiple sub-branches within the Gavilanes samples. This suggests an early stage of domestication characterized by high genetic variation, potentially reflecting local adaptations or a mixture of ancestral and derived traits. Bootstrap values, ranging from 0.84 to 0.96, provide statistical support for these clusters, with higher values indicating greater confidence in the distinct groupings. The observed diversity in *T. antiquorum* may also imply introgression or hybridization events, as suggested by intermediate samples from sites like Horbat Rosh Zayt and Narhan. Comparison with modern wheat species suggests that *T. antiquorum* may represent an early-stage domesticate or a semi-wild variety.

Conversely, *T. parvicoccum* displays a more compact clustering pattern, indicating a conserved lineage with reduced genetic variation. The tight grouping of Tell Batash samples suggests a uniform, domesticated population, consistent with later stages of domestication where selection pressures reduce diversity.

Intermediate samples (Figure 4), such as some from Punta de los Gavilanes and those from Horbat Rosh Zayt and Hala Sultan Tekke, are postulated as transitional forms, potentially linking the two primary taxa and contributing to the evolutionary trajectory of domesticated wheat.

Figure 5 displays carpological (caryopses) remains of small-grained wheat, from several archeological contexts of Punta de los Gavilanes. The various panels (A–G) highlight individual charred wheat grains and plant structures from different samples, showcasing variations in shape, preservation, and diagnostic features.

Seeds (Panels A–C, E,F, G): Samples 1814, 1807, and 2076_4 show charred wheat grains that appear small, suggesting an early domesticated or transitional wheat species. The small size of these grains is indicative of ancient cultivars that were in the initial stages of domestication or transitioning from wild to cultivated forms.

Some seeds exhibit surface cracking and deformation, due to charring processes during preservation. These alterations can occur when grains are exposed to elevated temperatures, which is common in archeological contexts where grains may have been accidentally or intentionally burned.

Morphological variation in size and roundness of wheat grains may indicate different taxa or different developmental stages. This variation suggests that multiple types of wheat were present, or that the grains were at various stages of maturity when preserved.

The charred preservation suggests these remains may have been exposed to fire or elevated temperatures, typical of archaeobotanical contexts. Charring is a common preservation method in archeological sites, as it can preserve organic materials that would otherwise decompose.

The analysis of the images reveals that the charred wheat grains and plant structures are from an early domesticated or transitional wheat variety. The small size of the grains and the presence of a dense-eared naked wheat fragment suggest that these remains are from a domesticated lineage. The charred preservation and morphological variations indicate that these grains were exposed to elevated temperatures and may represent different wheat taxa or developmental stages. Further microscopic analysis is required to confirm the specific taxonomic classification of these remains.

The sub image D (Sample 2074_3) in Figure 5 shows a charred plant fragment. This fragment belongs to a rachis of a dense-eared naked wheat, suggesting it may be related to *T. turgidum* subsp. *durum* (Desf.) Husn.*, T. sphaerococcum* Percival, or *T. aestivum* subsp. *compactum* (Host) Mac Key. The dense-eared characteristic is typical of domesticated wheat that has been selected for easier harvesting and higher yield. The preserved rachis internode and its structure are valuable for taxonomic classification, as these features differ between hulled and naked wheat. The absence of chaff (glumes) imprints on the internode supports the identification as naked wheat, which is free-threshing and does not require dehusking. If the rachis internodes were thinner or had remnants of chaff, it could suggest hulled domesticated wheat (e.g., Emmer, *T. turgidum* subsp*. dicoccum* (Schrank ex Schübl.) Thell., which has been widespread in Europe since the Neolithic period [97].

The charring process can cause fragmentation, making it harder to confirm whether it belonged to a wheat rachis (the main axis of the spike). However, its general structure and size are consistent with a tetraploid wheat species rather than a diploid (e.g., Einkorn, *T. monococcum*).

Dense-Eared (Compact Spike) Wheat Hypothesis: “Dense-eared” suggests that this wheat had tightly packed spikelets, which is a trait of free-threshing tetraploid wheat (e.g., some varieties of *T. turgidum* subsp*. durum*) [98]. If the sample indeed corresponds to rachis internodes of dense-eared wheat, it will mean that this wheat had undergone selection for compact spike structure, a feature seen in domesticated wheat adapted for easier harvesting. The sample belongs to a tetraploid wheat species (*T. turgidum* s.l.).

### 3.2. Probabilistic Allocation Análisis

The distribution of wheat taxa in archaeological samples from the Gavilanes and Almizaraque Argaric sites was analyzed using both Random Forest (RF) and logistic regression (LOG) methods. The findings reveal distinct patterns in the dominance and diversity of wheat taxa across the two sites (Table 2).

Table 2 presents a comprehensive taxonomic analysis of archaeobotanical wheat caryopses recovered from two Argaric archeological sites in southeastern Spain: Punta de Gavilanes in Murcia (*n* = 48) and Almizaraque in Almería (*n* = 517). The identification was conducted using two complementary approaches, the machine learning Random Forest and the logistic regression, with grains allocated to taxa in the left column of each type only when the probability exceeded 0.7, thereby ensuring robust taxonomic assignments and in the right column those allocated probability equal or inferior to 0.7.

The results demonstrate that *Triticum sphaerococcum* subsp. *sphaerococcum*, a hexaploid wheat, represents the predominant taxon at both sites, though with notably different proportions. At Punta de Gavilanes, this subspecies accounts for approximately 37–46% of the assemblage when considering high-confidence identifications from the Random Forest model (18 desiccated seeds) and logistic legression model (22 desiccated seeds). At Almizaraque, the number is even more pronounced, with 151 desiccated seeds identified with high confidence by Random Forest and 130 by logistic regression, but the dominance somewhat lesser, representing roughly 29–25% of the total assemblage, respectively.

The ploidy distribution across both sites reveals a marked preference for hexaploid wheats, which comprise the vast majority of identified specimens. Beyond *T. sphaerococcum* subsp. *sphaerococcum*, other hexaploid taxa include *T. sphaerococcum* subsp. *antiquorum*, *T. aestivum* subsp. *aestivum*, *T. aestivum* subsp. *compactum*, and *T. aestivum* var. *antiquorum*. Tetraploid wheats, primarily represented by various subspecies of *T. turgidum*, constitute a smaller but significant portion of the assemblages, particularly *T. turgidum* subsp. *parvicoccum* at Almizaraque. Diploid wheats are minimally represented, with only *T. monococcum* subsp. *monococcum* appearing in small numbers exclusively at Almizaraque.

The differential preservation states of the comparison material—desiccated, carbonized, and archaeobotanical—provide important insights for the model. Desiccated remains yielded the highest number of identifications for *T. sphaerococcum* subsp. *sphaerococcum* at both sites, while experimentally carbonized reference materials showed greater diversity in terms of species representation, particularly at Almizaraque where archeological grains coincided with reference carbonized samples including identifications of *T. aestivum* subsp. *aestivum*, *T. aestivum* subsp. *compactum*, and *T. monococcum* subsp. *monococcum*.

The substantially larger assemblage from Almizaraque permits more detailed observations regarding wheat diversity and agricultural practices during the Argaric period. The site exhibits considerably greater taxonomic diversity compared to Punta de Gavilanes, with representation across multiple ploidy levels and subspecies. This diversity may reflect either broader agricultural strategies, longer occupation periods, or different functional contexts of the sampled deposits.

Particular attention should be given to the two taxa identified exclusively within the archaeobotanical category: *T. turgidum* subsp. *parvicoccum* and *T. aestivum* var. *antiquorum*. The tetraploid *T. turgidum* subsp. *parvicoccum* shows substantial representation at both sites, with 70 high-confidence identifications by Random Forest at Almizaraque and smaller but consistent numbers at Punta de Gavilanes (6 high-confidence identifications), indicating its importance in the agricultural repertoire of these Argaric communities. Meanwhile, *T. aestivum* var. *antiquorum*, a hexaploid wheat, appears exclusively at Punta de Gavilanes in modest quantities (1 high-confidence identification by Random Forest), suggesting possible site-specific cultivation preferences or chronological variation in wheat species selection during the Argaric period.

The archaeobotanical assemblages reveal a notable absence or extremely limited representation of several wheat taxa that might otherwise be expected in Bronze Age Iberian contexts. Among the hexaploid wheats, *T. aestivum* subsp. *spelta* (L.) Thell. is entirely absent from both sites, while the tetraploid wheats demonstrate a similar pattern of scarcity. Specifically, *T. turgidum* subsp. *dicoccum* (Schrank ex Schübl.) Thell. and *T. turgidum* subsp. *durum* (Desf.) Husn. exhibit zero or negligible presence across the analyzed samples from both Punta de Gavilanes and Almizaraque. This distributional pattern suggests that these taxa were either not cultivated by Argaric communities in southeastern Spain or that their cultivation remained sufficiently marginal to escape detection in the archaeobotanical record, thereby highlighting the selective agricultural strategies employed during this period. Other missing taxa are a priori unexpected: *T. monococcum* subsp. *aegilopoides* (Link) Thell., *T. turgidum* subsp. *polonicum* (L.) Thell., *T. turgidum* subsp. *georgicum* (Dekapr. and Menabde) Mac Key ex Hanelt.

Conversely, the identification of *T. aestivum* subsp. *macha* (Dekapr. and Menabde) McKey, *T. turgidum* subsp. *dicoccoides* (Asch. and Graebn.) Thell., and *T. turgidum* subsp. *turanicum* (Jakubz.) Á.Löve and D.Löve proves particularly surprising and must be treated with considerable caution. The reliability of these identifications remains highly questionable given that each taxon is represented by only a single caryopsis within the entire assemblage. Such minimal representation raises important methodological concerns regarding the statistical confidence of these classifications and suggests that these identifications may reflect either the misclassification of artifacts inherent to the machine learning models or the presence of morphologically ambiguous specimens that fall at the boundaries of multiple taxonomic categories. Consequently, these singleton occurrences should not be interpreted as definitive evidence for the presence of these wheat subspecies in the Argaric agricultural system without corroborating evidence from additional specimens or independent lines of archaeobotanical inquiry.

The dimensions of the archeological wheat seeds are compared in Appendix A with those of the reference materials used for the training set. The archaeobotanical samples align well with their proposed modern counterparts in terms of morphometric ratios. For instance, *Triticum sphaerococcum* subsp. *sphaerococcum* shows remarkable consistency between ancient and modern specimens, with L/B ratios of 1.47 (Gavilanes) and 1.44 (Almizaraque) compared to 1.47 (modern desiccated). The relative shape characteristics (L/B, L/D, B/D ratios) tend to be more stable across time than absolute dimensions, suggesting these are reliable taxonomic markers that persist despite preservation effects.

Archeological grains consistently exhibit notable size reduction when compared to their modern counterparts, a phenomenon that appears across all samples examined. This reduction is particularly striking in *Triticum aestivum* subsp. *aestivum*, where specimens from Almizaraque demonstrate mean lengths of 4.78 mm compared to 6.30 mm in modern desiccated seeds, representing a substantial 24% decrease in overall length.

The preservation process itself introduces systematic morphological changes that significantly impact grain characteristics. Experimental carbonization of modern seeds reveals consistent patterns of dimensional alteration, with length measurements typically decreasing while breadth measurements often increase. For instance, *T. aestivum* subsp. *aestivum* shows a reduction from 6.30 mm to 5.53 mm in length following carbonization, fundamentally altering the proportional relationships that define grain morphology.

This differential preservation effect manifests most clearly in the breadth-to-length ratios of archeological specimens. Archeological samples consistently display elevated B/L values, thus lower L/B, when compared to modern desiccated seeds, indicates systematic shape distortion occurring during the preservation process. This pattern emerges across all examined taxa and represents a critical consideration for accurate taxonomic identification. Without proper acknowledgment of these preservation-induced morphological changes, researchers risk misclassifying archeological specimens based on distorted proportional characteristics that do not reflect the original grain morphology.

The patterns suggest that direct morphometric comparison between archeological and modern specimens requires careful calibration for preservation effects. The experimental carbonization data provides a valuable bridge, but more comprehensive taphonomic studies would strengthen identifications. Good correspondence in proportional measurements supports the continued use of these methods, but absolute size comparisons should be interpreted cautiously.

This dataset highlights the importance of using multiple reference standards (desiccated, carbonized, and archeological) when making taxonomic determinations in archaeobotanical contexts.

Figure 6 presents a correlation heatmap illustrating the relationships between the classification probabilities of various taxa in the identification of 2463 archeological wheat grains using a Random Forest model. This visualization highlights several significant taxonomic associations. The heatmap employs a blue-white-red color scale, where dark blue signifies strong positive correlations (approaching +1) (r > 0.6), white represents no correlation (0), and dark red indicates strong negative correlations (approaching −1) (r < −0.6). Taxa are arranged identically along both axes, which would have produced a symmetric matrix with perfect positive correlations (dark blue, value 1) along the diagonal. Since this is not informative the first column and the last row were excluded from the graphic.

Strong correlations between dry and carbonized specimens indicate preservation of diagnostic traits, as in *T. turgidum* subsp. *turgidum* and *T. sphaerococcum* subsp. *sphaerococcum*. By contrast, taxa such as *T. monococcum* show weaker correlations, reflecting distortion introduced by carbonization.

Within the hexaploid *T. aestivum* complex, subsp. *compactum* clusters with dry subsp. *aestivum* and *sphaerococcum* subsp. *antiquorum*, highlighting shared morphometric affinities among domesticated bread wheats. However, compactum diverges negatively from carbonized *sphaerococcum*, underscoring the need to account for preservation effects. Archeological *antiquorum* acts as a morphometric bridge between *sphaerococcum* and *parvicoccum*, suggesting that Iberian hexaploid wheats retained close affinities with lineages now confined to south and central Asia.

The *sphaerococcum* group demonstrates strong internal consistency across dry and carbonized forms and maintains positive correlations with both *antiquorum* and *parvicoccum*. These relationships confirm the phylogenetic significance of *sphaerococcum*-like finds in Iberian assemblages and their link to Asian wheat lineages.

The tetraploid *T. turgidum* subsp. *parvicoccum* [44,80] occupies an unusual position, correlating positively with *antiquorum*, *monococcum aegilopoides*, and dry *monococcum* while diverging from core tetraploids such as *turgidum*, *spelta*, and *carthlicum*. This distinctiveness suggests that *parvicoccum* represents a relic lineage aligned with peripheral or “lost” morphotypes rather than mainstream domesticated wheats.

Overall, the correlation structure supports a phylogenetic model in which diploids connect tightly to transitional tetraploids, while hexaploids diverge more strongly, with *antiquorum* and *sphaerococcum* bridging these groups. The absence of *antiquorum* [46,47,48,49] and *parvicoccum* [37,40,41,43,44] in modern western Europe likely reflects the extinction of distinctive lineages once integral to regional agriculture. These findings emphasize both the importance of experimental carbonization datasets for robust archaeobotanical interpretation and the potential of morphometric approaches to uncover lost chapters of wheat evolutionary history.

The overarching phylogenetic message revealed by this correlation matrix demonstrates that Iberian archeological wheats, represented by *antiquorum* and *parvicoccum*, were not marginal outliers within the wheat evolutionary framework but were instead integral components of broader evolutionary lineages, particularly those connecting to *sphaerococcum* and primitive tetraploid forms. Their conspicuous absence from modern western European wheat diversity reflects the local extinction of distinctive genetic lineages that once contributed to regional agricultural systems. This finding underscores how archaeobotanical morphometric approaches can illuminate previously hidden chapters of wheat phylogenetic history, revealing the extent of genetic diversity that has been lost through millennia of agricultural intensification and crop standardization.

In terms of site comparison, Gavilanes exhibits a higher concentration of *T. sphaerococcum* (Dry), while Almizaraque displays greater taxonomic diversity, with more significant presence of *T. aestivum* subspecies. This analysis suggests that *T. sphaerococcum* was the predominant wheat cultivated at both archeological sites, with greater taxonomic diversity at Almizaraque, potentially indicating different agricultural practices or environmental conditions between the two settlements (Table 2).

### 3.3. Comparative Assessment of Classification Techniques

The methodological comparison between RF and LOG reveals notable differences in predictions for certain taxa. For example, *T. aestivum* subsp. *aestivum* (Dry) at Almizaraque shows a 20.46% presence by RF but 34.17% by LOG. The mean probability values align with the percentage distributions but exhibit some variations, indicating different confidence levels in the identifications.

The comparative analysis, presented in Table 3, illuminates differences and coincidences in wheat grain identification from the Punta de los Gavilanes archeological site, using contemporary machine learning techniques, specifically Random Forest, and of logistic regression algorithms.

Machine learning analyses demonstrate a clear taxonomic preference for *T. sphaerococcum* (Dry), which emerges as the predominant species across most samples examined by both algorithms. This identification pattern persists regardless of initial classical determinations, with algorithms frequently reclassifying grains originally designated as *Ta* (*T. aestivum* subsp. *compactum* var. *antiquorum*) or categorized as “Doubtful” into the *T. sphaerococcum* classification. Such consistent reclassification patterns underscore the analytical strength of machine learning approaches in recognizing morphological characteristics associated with *T. sphaerococcum* as the primary wheat species present at the site.

The temporal scope of these samples encompasses approximately four centuries of Bronze Age occupation, spanning from 2290 BC to 1890 BC. Throughout this extended chronological framework, the persistent identification of *T. sphaerococcum* suggests remarkably stable cultivation practices at Punta de los Gavilanes. This continuity indicates that *T. sphaerococcum* represented a dependable and favored wheat variety among the site’s inhabitants during this historical period, reflecting established agricultural traditions that endured across multiple generations.

The methodological comparison reveals striking concordance between Random Forest and logistic regression predictions, with both approaches identifying *T. sphaerococcum* as the most probable taxon in eleven of thirteen samples examined (twenty grains out of twenty-six). This consistency reinforces confidence in the machine learning results while highlighting the reliability of both analytical approaches. However, Sample 3077 presents a notable exception, with Random Forest suggesting *T. aestivum* subsp. *compactum* while logistic regression identifies *T. aestivum* subsp. *aestivum*. This methodological divergence emphasizes the importance of investigating the specific morphological characteristics that distinguish this particular specimen.

The remarkable agreement between Random Forest and logistic regression for Punta de los Gavilanes wheat grains results strengthen confidence in these determinations while suggesting that *T. sphaerococcum* constituted the primary wheat species at Punta de los Gavilanes throughout the Bronze Age occupation period. These findings highlight the transformative potential of machine learning applications in archeological grain identification, offering enhanced accuracy and reliability while providing valuable insights into ancient agricultural practices and crop selection strategies.

The comparative analysis of Table 4 examines the relationship between historical taxonomic identifications established by Tellez and Ciferri [68] and two contemporary machine learning methodologies applied to wheat grain samples from the Almizaraque archeological site, dating to approximately 2000 BC. This investigation reveals significant divergences between traditional and algorithmic classification approaches that fundamentally challenge previous understanding of the site’s agricultural composition.

Machine learning and logistic regression analyses present a substantial taxonomic reassessment of the Almizaraque assemblage. While Tellez and Ciferri [68] originally classified the samples as *T. aestivum* subsp. *aestivum* or *T. aestivum* subsp. *compactum*, algorithmic approaches frequently reclassified these specimens as *T. sphaerococcum*. Among the eleven samples examined, six received *T. sphaerococcum* identifications from at least one of both methods, despite none having been originally classified within this taxon. This systematic reclassification pattern demonstrates the capacity of machine learning and logistic regression techniques to detect morphological characteristics and taxonomic relationships that may have been overlooked or misinterpreted through traditional analytical approaches.

The methodological comparison between Random Forest, a machine learning method, and logistic regression reveals moderate concordance, with both approaches reaching agreement in seven of eleven samples, representing 64% consensus. When methodological disagreements occurred, they typically followed a pattern where Random Forest identified specimens as *T. aestivum* subsp. *compactum* while logistic regression classified them as *T. sphaerococcum* or vice versa. These discrepancies illuminate the nuanced differences between machine learning algorithms and logistic regression and underscore the inherent complexity involved in archeological taxonomic identification, particularly when working with morphologically variable ancient material.

The machine learning and logistic regression analyses reveal notably greater taxonomic diversity at Almizaraque compared to the patterns observed at Punta de los Gavilanes. While *T. sphaerococcum* emerges as a prominent component, both *T. aestivum* subsp. *aestivum* and *T. aestivum* subsp. *compactum* maintain substantial representation within the reassessed assemblage. This enhanced diversity suggests a more complex agricultural landscape than previously recognized, potentially indicating the concurrent cultivation of multiple wheat species or varieties that reflect sophisticated agricultural management strategies employed by the site’s inhabitants.

Several samples demonstrate complete taxonomic reclassification that highlights the analytical power of machine learning approaches but also of logistic regression. Sample “Casa 32 Caja 1857. Muestra 4,” originally identified as *T. aestivum* subsp. *compactum*, received unanimous reclassification as *T. sphaerococcum* from both algorithmic methods. This unanimous reassignment demonstrates the capacity of modern techniques to identify consistent taxonomic patterns that traditional morphological analysis may not have recognized or adequately weighted.

The comprehensive analysis demonstrates that machine learning and logistic regression methodologies provide substantial reinterpretation of historical taxonomic identifications, suggesting that *T. sphaerococcum* maintained a significantly more prominent presence at Almizaraque than previously recognized through traditional analytical approaches. However, the moderate probability ranges observed throughout the dataset indicate persistent uncertainty in taxonomic assignments, which may reflect authentic morphological variation within ancient wheat populations or alternatively represent inherent limitations within current identification methodologies. This investigation emphasizes the transformative value of machine learning applications and of logistic regression in archeological grain identification, offering enhanced analytical precision while revealing new insights into ancient agricultural practices and the complexity of cultivated wheat diversity in prehistoric contexts.

## 4. Discussion

### 4.1. Interpretation of Taxonomic Results

#### 4.1.1. What Represents the Presence of Sphaerococcum-Type Wheats in These Iberian Archeological Contexts?

Considering the results of the identification and correlation analyses and given that *T. sphaerococcum* is presently an Asian endemic species [59,60,61,62,81,99] (Figure 7) and that the Almizaraque and Gavilanes samples date back to around 2000 BC, several alternative interpretations merit careful consideration.

The presence of *sphaerococcum*-type wheats in these Iberian archeological contexts could reflect a historically broader distribution than the one that exists today. It is entirely possible that *T. sphaerococcum* enjoyed a wider geographical range during the third millennium BC, extending well beyond its current restriction to India and central Asia. Environmental changes over the subsequent four millennia, including significant climatic shifts and intensifying human activities, could have progressively contracted its distribution to the geographically limited range observed in contemporary agriculture. Such range contractions are well-documented phenomena in crop species, where historical distributions often exceeded modern cultivation areas due to subsequent environmental pressures and changing agricultural systems.

Human migration patterns and ancient trade networks provide another compelling explanation for the presence of these wheat forms in prehistoric Iberia. Archeological evidence increasingly demonstrates that sophisticated trade routes connected the Indian subcontinent and central Asia with the Mediterranean basin during the Bronze Age. These networks could have facilitated the long-distance movement of seeds and grains, enabling the cultivation of *T. sphaerococcum* in regions far removed from its primary center of origin and domestication. Such agricultural exchanges were integral components of early trade relationships, with exotic crops often serving both practical and prestige functions in recipient societies. The center of origin of hexaploid wheat is central Asia, and *T. sphaerococcum* is a hexaploid wheat, while the Fertile Crescent is the center of origin of diploid and tetraploid wheats [59].

The possibility of taxonomic complications cannot be dismissed, particularly given the morphological complexity within hexaploid wheat taxa. The archeological specimens identified as *T. sphaerococcum* might represent closely related forms that were more prevalent in the Iberian Peninsula during the Bronze Age but have since disappeared from the archeological and historical record. Morphological convergence between distinct wheat lineages can create identification challenges, especially when working with carbonized archeological remains that may have undergone significant taphonomic alteration. This scenario would suggest the existence of now-extinct hexaploid wheat varieties that shared key morphometric characteristics with Asian *sphaerococcum* but represented independent evolutionary lineages.

Environmental adaptation capabilities may have differed significantly between ancient and modern wheat populations. The *sphaerococcum* lineage might have possessed greater phenotypic plasticity in the past, allowing successful cultivation across diverse climatic and edaphic conditions that would challenge contemporary varieties. Subsequent changes in agricultural practices, including intensification of cultivation methods and standardization of crop varieties, combined with environmental factors such as climate change and soil degradation, could have reduced the adaptive flexibility of these wheat forms and restricted their viable cultivation range. The cultivation of *T. sphaerococcum* declined significantly as other wheat species, offering likely higher yields and better qualities for bread and other foods, became increasingly preferred.

The genetic diversity within *T. sphaerococcum* was greater in antiquity than what survives today. Ancient populations may have encompassed numerous varieties or subspecies adapted to specific regional conditions, representing a much broader genetic base than current Asian populations. Over the millennia, many of these locally adapted forms could have succumbed to extinction through genetic erosion, environmental pressures, and the progressive replacement of diverse local varieties with standardized, high-yielding cultivars. This genetic contraction would have eliminated variants that were once successful in the Mediterranean and European environments.

Cultural preferences and agricultural traditions provide additional explanatory frameworks for understanding the historical distribution of *sphaerococcum*-type wheats. Ancient Iberian societies might have deliberately cultivated specific wheat varieties based on particular culinary requirements, storage properties, or symbolic significance that differed markedly from modern agricultural priorities. Cultural practices surrounding food preparation, ritual usage, and trade relationships can profoundly influence which crop varieties are maintained within agricultural systems. The subsequent shift in cultural preferences, potentially accelerated by contact with other agricultural traditions or changes in social organization, could have led to the abandonment of these specialized wheat forms in favor of varieties that better suited evolving cultural and economic needs.

These various interpretations are not mutually exclusive; they operated in combination to produce the complex biogeographical patterns observed in the archeological record. The identification of *sphaerococcum*-type wheats in Bronze Age Iberia thus illuminates the dynamic nature of ancient agricultural systems and highlights how contemporary crop distributions represent only a fraction of the diversity that once characterized global wheat cultivation.

These interpretations highlight the complex interplay between historical, environmental, and cultural factors that shape the distribution of plant species over time. Further research, including genetic analysis of the archeological samples, could provide more insights into the historical distribution and evolution of *T. sphaerococcum*.

#### 4.1.2. What and Where Are *T. parvicoccum* and *T. antiquorum?*

According to the IFPNI International Editorial Board [20], *Triticum parvicoccum* Kislev is a validly published name for an archeological macro-fossil species of wheat from the Holocene. However, it is debatable whether archaeobotanical remains can be considered true fossils. Additionally, Rivera et al. [24] have demonstrated that archaeobotanical taxa are not fossils. Moreover, the publication of a novel taxon in English in 1980, in the case of being modern, did not fulfill the requirements for valid publication as established by the International Code of Botanical Nomenclature (Shenzhen Code). In reconstructing the ear morphology of this ancient small-grain wheat, Kislev in 2009 [44] classified it under *T. turgidum* as *T. turgidum* subsp. *parvicoccum*. Nevertheless, this classification does not align with the rules of botanical nomenclature [23]. Finally, Kislev validated this combination in 2025 [24].

This subspecies is considered the oldest naked wheat [37,42,43,44,59] and was first reported in the Near East during the Pre-Pottery Neolithic B (PPNB) period. Kislev [37] documented a series of finds from levels dating from the 6th millennium to the 4th century BC at sites in west Asia, the Balkans, and Cyprus. Feldman and Kislev [41] describe *T. parvicoccum* as an extinct subspecies of tetraploid *T. turgidum*, characterized by its free threshability, ear compactness, and small grain size. Its earliest records were found in the PPNB phase II of Tell Aswad (8900–8600 BP) and the submerged PPNC of Atlit-Yam (9200–8500 cal. BP). It remained one of the most important crops in the Near East until the Hellenistic period.

*T. turgidum* subsp. *parvicoccum* is distinguished by its noticeably short internodes and small grains, which are oval to elliptic [37] (Appendix A). It possesses characteristically smaller caryopses than other wheat species. Some authors recognize these small-grained wheat forms as mere ear-type variants within *T. turgidum* [39,65]. These dense-eared tetraploids appear to parallel the variation found in hexaploid “compactums” [100,101].

Kislev in 2009 [43] provided a list of seven diagnostic differential characters, six of which are strictly morphological and they distinguish *T. parvicoccum* of the Near East and the Balkans from hexaploid wheats. These characteristics are included in Appendix A.

This subspecies (*T. turgidum* subsp. *parvicoccum*) is an early example of free-threshing wheat [102] and is believed to be an ancestor of free-threshing tetraploid wheats such as *T. turgidum* subsp. *durum* [103]. Schultze-Motel [21] reported its presence in Georgia (Transcaucasia) from Vani (Imeretija, 3rd–1st centuries BC) and Senako (Amchetskjj, 12th century AD). Liu et al. [81] noted the presence of small-grained free-threshing wheats in central Asia (Kazakhstan), India, Pakistan, and China, associated with the eastward spread of wheat during the third and second millennia BC. The dimensions of these wheat samples fall within the range of *T. turgidum* subsp. *parvicoccum*. Liu et al. [81] inferred that these traits were not indicative of ancestry but were a direct result of active crop selection to suit culinary needs.

Regarding the ancestral material of hexaploid free-threshing wheat designated as *T. vulgare* var. *antiquorum* Heer (*T. aestivum* subsp. *compactum* var. *antiquorum* (Heer) H.Messik.), Oswald von Heer in 1865 [25] described this taxon originally from archeological grain remains discovered in Switzerland. The presumed extinct wheat taxon resurfaced when morphologically similar plants were collected in 1965 and described in 1982 from the western Pamirs region of Tajikistan. However, the inherent limitations in comparing archeological specimens with extant plants prevent definitive confirmation that these modern samples represent the same ancient taxon.

Following examination of this newly collected material, Udachin proposed elevating von Heer’s variety to species rank as *T. antiquorum* (Heer) Udachin [46,47].

When Flaksberger reclassified *T. vulgare* var. *antiquorum* as *T. compactum* var. *antiquorum* (Heer) Flaksb., he emphasized its close resemblance to hexaploid wheats based on glume morphology and spike density, while noting its distinction from typical *T. compactum* primarily in its spherical grains [48].

Both the Swiss archaeological specimens and the Pamir living material exhibit characteristic features of the “inflatum” wheat type (Appendix A): awnless dense spikes with prominently developed spikelet keels extending to their bases. At the keel’s base appears an indentation typical of hexaploid wheat. The keels and scale bases display similar longitudinal wrinkling. Each spikelet contains 3–4 rounded grains. These morphological correspondences between archeological and contemporary samples highlight their potential taxonomic relationship while acknowledging the challenges in establishing definitive identity across millennia.

#### 4.1.3. The Significance of Small-Grained Free-Threshing Wheat at Punta de los Gavilanes

The archaeobotanical analysis of Punta de los Gavilanes yielded 2643 identifiable carpological remains and 500 fragments from the site. Annual crops dominated the assemblage, particularly cereals and pulses, with *Hordeum vulgare* L. (1865 grains) and small-grained free-threshing wheat (478 grains) representing the most abundant taxa. Additional crops were present in lesser quantities, including *Vicia faba* L. and *Lathyrus sativus* L. The assemblage also contained associated field weeds alongside halophytic and subhalophytic plant species. The relative abundance of cereal species demonstrates a distinctive distributional pattern across the site.

The image in Figure 8 presents a comprehensive chronological analysis of cereal crop distribution across multiple archeological phases at Punta de los Gavilanes, arranged sequentially from oldest (left) to most recent (right). This diachronic representation reveals significant temporal shifts in agricultural practices and crop preferences at the site.

The earliest phase (IVa1 TM) exhibits remarkable taxonomic diversity dominated by *T. sphaerococcum*, notably subsp. *sphaerococcum* (cf. Figure 9) but also subsp. *antiquorum*, (cf. Figure 9 for modern reference material) with supplementary presence of *T. turgidum* subsp. *parvicoccum* and *T. aestivum* subsp. *compactum* (cf. Figure 9). This suggests an initial agricultural strategy focused on diverse wheat cultivation, with minimal barley presence (only three caryopses of *Hordeum vulgare*).

A dramatic transition occurs in phase IVa2 TM_TS, marked by a substantial reduction in wheat diversity and abundance, with *T. sphaerococcum* declining. Simultaneously, barley cultivation (*Hordeum vulgare* and *H. distichon (=syn. H. vulgare* ssp. *distichon* (L.) Körn.)) rises significantly, constituting approximately 75% of the assemblage. The presence of *T. turgidum* subsp. *parvicoccum* during this phase is also noteworthy.

Subsequent phases (IVb through II TS_TM) demonstrate a continued and intensified preference for *H. vulgare*, which progressively dominates the assemblage, reaching near-complete monopoly with 1938 caryopses by phase IVd TSM. The overall wheat presence becomes increasingly marginal, with occasional minor occurrences of *T. sphaerococcum*, *T. turgidum* subsp. *parvicoccum*, and *T. aestivum* subsp. *compactum*.

This chronological progression reveals a significant agricultural transformation from wheat-dominant to barley-dominant cultivation strategies, potentially reflecting adaptations to changing environmental conditions, socioeconomic preferences, or technological innovations in Bronze Age southeastern Iberia.

Of particular significance is the small-grained free-threshing wheat, which merits detailed examination. Morphometric analysis of the recovered grains revealed dimensions of 5.11 ± 0.69 mm (length), 3.2 ± 0.38 mm (breadth), and 3.1 ± 0.36 mm (thickness), with a length/breadth ratio of 1.58 ± 0.17. These measurements partially align with dimensional ranges previously established by Kislev in 1980 [37] for “*T. parvicoccum*” as follows: 4.53 ± 0.68 mm (length), 2.8 ± 0.45 mm (breadth), and 2.42 ± 0.39 mm (thickness), with a length/breadth ratio of 1.62 ± 0.17. Kislev et al. in 2006 [40] subsequently refined the taxonomic concept of “*T. parvicoccum*” based on an extensive assemblage from Tell Batash comprising over 100,000 grains and 100 rachis fragments, characterized by more elongated and slender grains: 5.35 ± 0.44 mm (length), 2.8 ± 0.3 mm (breadth), and 2.37 ± 0.27 mm (thickness), with a length/breadth ratio of 1.9 ± 0.18. We used this last ensemble as a reference for the taxon for the multivariate, logistic regression and machine learning analyses.

The recovered grains from Gavilane exhibit distinctive morphological characteristics: short, oval to elliptical in outline, with maximum width at the middle or occasionally in the lower third, and maximum thickness in the lower third. The Punta de los Gavilanes specimens are notably wider and thicker than the average metrics reported by Kislev [37]. Rachis fragments (Figure 4) display breakage at non-specific points of the rachis internode with glumes unattached to the node, suggesting that this small-grained wheat represents a naked type consistent with Kislev’s [37] description. A single well-preserved rachis fragment allowed measurement of an internode length of 3.7 mm (Figure 5). The presence of short internodes characteristic of the tetraploid free-threshing type, comparable to those documented by Kislev (1980, 2009), further supports identification as small-grained free-threshing wheat.

However, the prevailing presence of *T. sphaerococcum* subsp. *sphaerococcum* among wheats in Punta de los Gavilanes and in Almizaraque takes us to the theory proposed by Udachin and Goncharov [46,47,49], which suggests a connection between *T. sphaerococcum* and *T. vulgare* var. *antiquorum*. This theory is explored in a study that addresses the relationship between archeological materials and potentially analogous living populations of wheat. The study highlights the challenges in definitively establishing the identity between archeological and contemporary materials, proposing an intermediate approach to understand their connection [49].

The specimen used in the study, which included complete plants with leaves and inflorescences, was identified as ‘*T. antiquorum’* [46,47]. This identification was based on the traditional concept of *T. vulgare* var. *antiquorum* [48] but recognizing significant differences between a compactum-like primitive wheat and the *Triticum sphaerococcum* Percival group. The study also notes that the original herbarium sheet for this specimen is missing, and further search for it is not possible due to the passing of Professor Udachin. However, N.P. Goncharov planted seeds from the original accession, which were provided by Udachin as an authentic specimen of “*T. antiquorum.*” [49].

The theory suggests that *T. sphaerococcum* subsp. *antiquorum* represents a subsample from the original accession, which was provided to N.P. Goncharov by Professor Udachin. This subspecies is described as having a compactum-like primitive wheat morphology, distinct from the typical *T. sphaerococcum* Percival (Figure 9) [49] (Appendix A).

In summary, the theory proposed by Udachin and Goncharov suggests a historical and taxonomic connection between *T. sphaerococcum* and *T. vulgare* var. *antiquorum*, supported by the analysis of archeological materials and living populations. This connection is crucial for understanding the origins and evolution of hexaploid wheat but also for understanding our surprising results from southeastern Iberian Peninsula.

The dimensional heterogeneity observed in the small-grained free-threshing wheat assemblage from Punta de los Gavilanes was expected. Kislev and Melamed [86] reported considerable variation in average dimensions between samples of “*T. parvicoccum*”, ranging from 3.8 × 2.6 × 2.3 mm to 5.3 × 3.1 × 2.7 mm. This variability reflects a combination of genetic diversity and environmental factors, as the archeological context may contain remains from different cultivation years or fields. It is well established that cereal grain dimensions fluctuate in response to annual climatic conditions and soil fertility [48]. While this pattern of variation describes the Punta de los Gavilanes assemblage, several samples with more rounded grains may represent hexaploid free-threshing wheats of the “sphaerococcum” or “compactum” type.

Plant remains were recovered throughout all analyzed phases and from various contexts within the site; however, small-grained free-threshing wheat remains were exclusively confined to Phases IVa-IVc. Notably, over 99% of the small-grained free-threshing wheat specimens were recovered from House 1TM (samples 2074-23, 2076-2,-3,-4,-6, and 2077-1,-23), 3700 ± 30 BP, specifically from a small cereal roasting oven near the dwelling entrance, an adjacent milling area, the domestic hearth, and floor deposits (Figure 3A). This restricted distribution is remarkable, as imported plant species that failed to establish persistently are rarely documented archeologically. The disappearance of small-grained free-threshing wheat may reflect displacement by more productive cereals or association with a specific colonization event, wherein migrants introduced these seeds as part of their cultural repertoire before being subsequently displaced by local or foreign populations (along with their cultivars).

Small-grained free-threshing wheat predominates in the earliest Phase IVa (Figure 8), with its presence in later phases becoming merely vestigial. The esparto basket from Phase IVb (Argaric Early Bronze Age period) (3560 ± 25 to 3425 ± 30 BP/3860 to 3623 cal. y. BP) (Figure 3B) (sample 1451) provides valuable insight into the environmental conditions under which cultivation occurred, as this container accumulated diverse carpological remains representing various ecological habitats. The botanical assemblage includes crops such as *Hordeum vulgare* and small-grained free-threshing wheat alongside halophytic species like *Salicornia lagascae* (Fuente, Rufo, and Sánchez Mata) Piirainen and G.Kadereit and subhalophytes including *Polygonum aviculare* L. The proportionally lower representation of *H. vulgare* and small-grained free-threshing wheat seeds compared to halophytic species suggests that harvested grains had already been removed from the basket prior to carbonization, with the remaining specimens representing processing waste.

#### 4.1.4. *T. parvicoccum* and *T. antiquorum* in the Iberian Peninsula

According to Kislev [37], the small-grained tetraploid naked wheat, *T. parvicoccum*, expanded from the Near East to the western Mediterranean basin and was present during the late Neolithic in Nerja, Spain [104]. The presence of *T. parvicoccum* has been reported under various names from the Chalcolithic site of Cova de l’Or (Alicante) and the Bronze Age sites of Almizaraque (Almería) and Cueva de Nerja (Málaga) [45,104,105]. Notably, these three sites are situated within the Thermomediterranean and Mesomediterranean climatic zones, ranging from 0.5 to 20 km from the seashore and at altitudes between 100 and 650 m above sea level. In comparison, Punta de los Gavilanes is more exposed to soil salinity and sea winds laden with moisture and salt.

Rovira (2007) [106] documented numerous wheat remains classified as *T.*
*compactum* from Neolithic, Chalcolithic, and Bronze Age contexts at various sites across the southeastern Iberian Peninsula. The caryopsis morphology, characterized by small grains, is consistent with both tetraploid and hexaploid wheat forms. However, in the absence of diagnostic ear rachis fragments (internodes) or a sufficiently broad comparative collection, it remains difficult to determine whether these caryopses derive from tetraploid wheats or hexaploid *compactum* varieties [107]. This taxonomic ambiguity is exemplified by the single caryopsis recovered from the Chalcolithic site of Las Pilas [108]. Nevertheless, the application of this novel technique may provide means to refine the identification of such archeological wheat remains and to evaluate the extent to which they correspond to similar types identified at Punta de los Gavilanes and Almizaraque.

### 4.2. Implications for Current Wheat Conservation, Breeding Programs

The identification of phylogenetically distinct wheat taxa—such as *Triticum sphaerococcum* and related lineages now restricted to central and south Asia—in ancient western European assemblages reveals substantial genetic erosion over the past 4000 years, highlighting that modern European wheat germplasm represents only a fraction of the cultivated diversity that once existed. The detection of *T. turgidum* subsp. *parvicoccum* and *T. aestivum* var. *antiquorum* in these ancient assemblages raises critical questions about whether closely related lineages still survive in marginal agricultural systems or traditional farming communities, particularly in regions of west Asia in the first case and central and south Asia in the second, where phylogenetically similar wheats persist today. This phylogenetic turnover and regional extirpation of entire lineages underscore the urgency of conserving extant wheat diversity in western, central, and south Asian gene banks, as these populations may harbor the last remnants of ancient cultivated gene pools that were once geographically widespread, and systematic surveys of traditional landraces could potentially recover living relatives of these archaeobotanically documented taxa. For breeding programs, this evidence of cryptic phylogenetic diversity suggests that traits and adaptive alleles present in these historically widespread but now geographically confined lineages—including potential survivors of *T. turgidum* subsp. *parvicoccum* and *T. aestivum* var. *antiquorum* or their close relatives—could offer valuable genetic resources for crop improvement, potentially including characteristics that enabled their successful cultivation across diverse Mediterranean and European environments during the Bronze Age. The study’s demonstration of long-term genetic erosion in domesticated wheat populations parallels conservation concerns typically associated with wild species, emphasizing that cultivated plant diversity requires proactive preservation strategies to prevent further lineage loss and maintain options for future agricultural resilience in the face of climate change and evolving pest pressures.

### 4.3. Limitations

The machine learning classifiers used in this study were trained on modern reference specimens (848 dry and 340 carbonized grains) and a limited number of archaeobotanical type samples (15 *T. turgidum* subsp. *parvicoccum* and 38 *T. aestivum* var. *antiquorum*). As a result, the training data may not fully capture the morphological diversity of ancient wheat lineages, some of which are now extinct or regionally lost. Expanding the modern training dataset to include a wider range of germplasm—particularly from central and south Asia, where related ancient lineages still persist—would likely improve model performance and taxonomic coverage. Increasing the representation of type material would also be valuable; for instance, *T. turgidum* subsp. *parvicoccum* is well preserved and accessible in the Bar-Ilan University archaeobotanical collections from Tell Batash. In contrast, expanding representation *of T. aestivum* var. *antiquorum* remains difficult, as the key carbonized reference samples were lost during a 20th-century loan from the Zurich museum to Russia.

## 5. Conclusions

This morphometric analysis of archeological wheat remains from Almizaraque and Punta de los Gavilanes (ca. 4000 BP) fundamentally challenges current understanding of *Triticum* paleogeography and phylogenetic distribution during the Bronze Age. Applying Random Forest and logistic regression algorithms to comprehensive morphometric datasets comparing archeological specimens with modern desiccated and experimentally carbonized reference materials across diploid, tetraploid, and hexaploid wheat taxa, we identified taxonomic compositions that significantly expand the known geographic and temporal distribution of several critical wheat lineages.

The identification of *Triticum sphaerococcum* subsp. *sphaerococcum* and subsp. *antiquorum* within these assemblages represents a paradigmatic shift in understanding early wheat cultivation in the Iberian Peninsula. These taxa, currently restricted to central Asian agricultural systems, demonstrate a far broader historical distribution than previously recognized, suggesting either extensive prehistoric trade networks facilitating long-distance crop dispersal or evolutionary origins across a wider geographic range than contemporary distributions indicate. Consistent identification through both analytical approaches across multiple samples and chronological contexts at both sites indicates established cultivation practices rather than isolated introductions, implying sustained agricultural integration of these central Asian wheat forms within Bronze Age Iberian farming systems.

The archaeobotanical documentation of *T. turgidum* subsp. *parvicoccum* further expands the known prehistoric distribution of tetraploid wheat diversity in western Europe, contributing to growing evidence for greater morphological and genetic diversity within early European wheat cultivation than traditionally recognized. The presence of this subspecies alongside *T. sphaerococcum* varieties indicates sophisticated agricultural management strategies incorporating diverse wheat lineages with potentially different adaptive advantages and culinary applications.

Methodologically, this research establishes machine learning and logistic regression as transformative tools for archaeobotanical taxonomy, offering unprecedented precision in distinguishing morphologically similar taxa while accounting for preservation-induced alterations. Systematic comparison of archeological specimens with both modern desiccated and experimentally carbonized reference materials provides crucial insights into taphonomic processes affecting grain morphology, enabling more accurate taxonomic assignments. The remarkable consistency between Random Forest and logistic regression results across samples strengthens confidence in these identifications while highlighting the robustness of both approaches for complex morphometric datasets.

From a phylogenetic perspective, these findings necessitate reconsideration of wheat evolutionary trajectories and subspecific diversification timelines. The presence of *T. sphaerococcum* lineages in Bronze Age Iberia suggests either earlier diversification than previously recognized or more extensive prehistoric distribution patterns that subsequently contracted to central Asian refugia. These data contribute to the evidence that wheat phylogeographic patterns reflect complex historical processes involving multiple expansion, contraction, and recolonization events shaping contemporary distributions.

Beyond taxonomic identification, these findings illuminate prehistoric agricultural networks, crop dispersal mechanisms, and cultural exchange systems facilitating wheat variety movement across vast distances during the Bronze Age. The sustained cultivation of central Asian wheat varieties in Iberian contexts implies sophisticated knowledge transfer systems and suggests Bronze Age agricultural communities actively maintained diverse crop portfolios through extensive trade or migration networks.

Future research should prioritize ancient DNA analysis to confirm taxonomic identifications and investigate phylogenetic relationships with contemporary wheat populations. Expanded morphometric analyses incorporating larger sample sizes from diverse geographic contexts will further refine understanding of prehistoric wheat diversity patterns. Integration of machine learning approaches with genomic data promises to revolutionize archaeobotanical research, providing unprecedented insights into crop evolution, human migration patterns, and co-evolutionary dynamics between agricultural communities and their cultivated resources.

This study demonstrates that Bronze Age agricultural systems were more complex and geographically connected than previously recognized, fundamentally altering understanding of early wheat cultivation patterns and their role in shaping human societies and crop evolutionary trajectories. The identification of central Asian wheat varieties in prehistoric Iberian contexts represents a significant contribution to *Triticum* phylogeography while establishing machine learning morphometrics and logistic regression as essential tools for advancing archaeobotanical research and crop evolutionary history. Ultimately, our results confirm the hypothesis proposed by R. A. Udachin over sixty years ago regarding the presence of wheat varieties related to *T. sphaerococcum* in western European prehistory.

## Figures and Tables

**Figure 1 genes-16-01477-f001:**
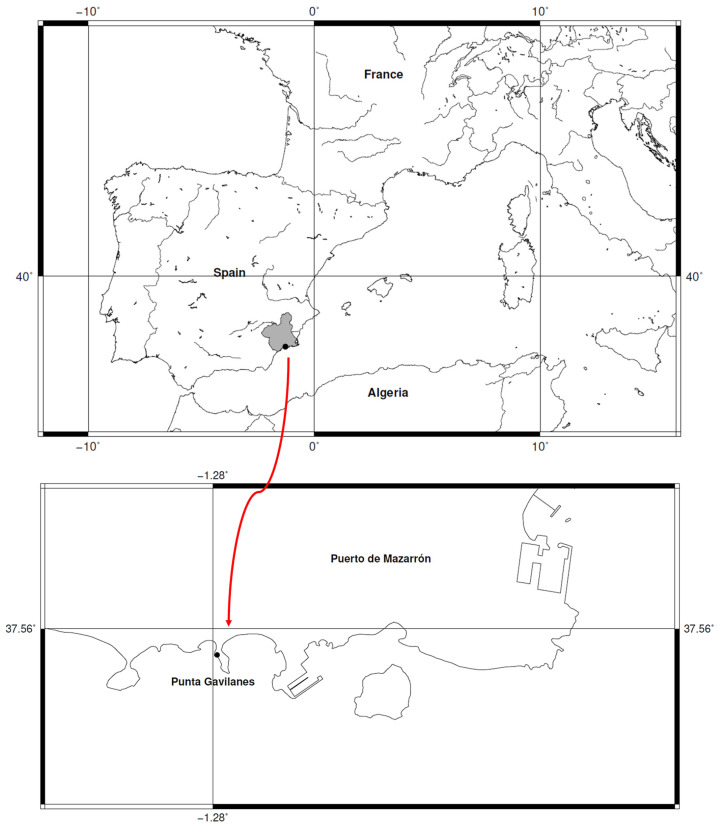
Geographic location of the Punta de los Gavilanes archeological site situated within Mazarrón Bay, southeastern Iberian Peninsula. Map: Francisco Alcaraz.

**Figure 2 genes-16-01477-f002:**
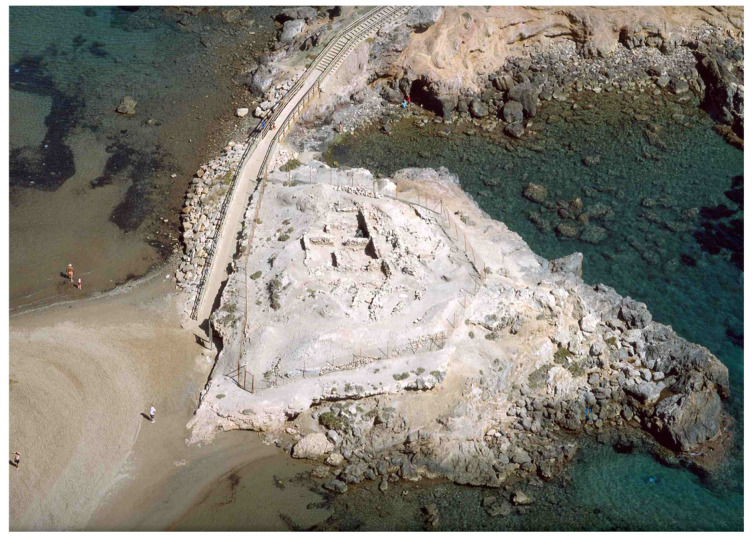
Aerial view of Punta de Los Gavilanes coastal settlement. This settlement occupied a strategic coastal promontory, facilitating maritime connections while providing access to inland agricultural territories. Source: Gavilanes Archeological Project Archive.

**Figure 3 genes-16-01477-f003:**
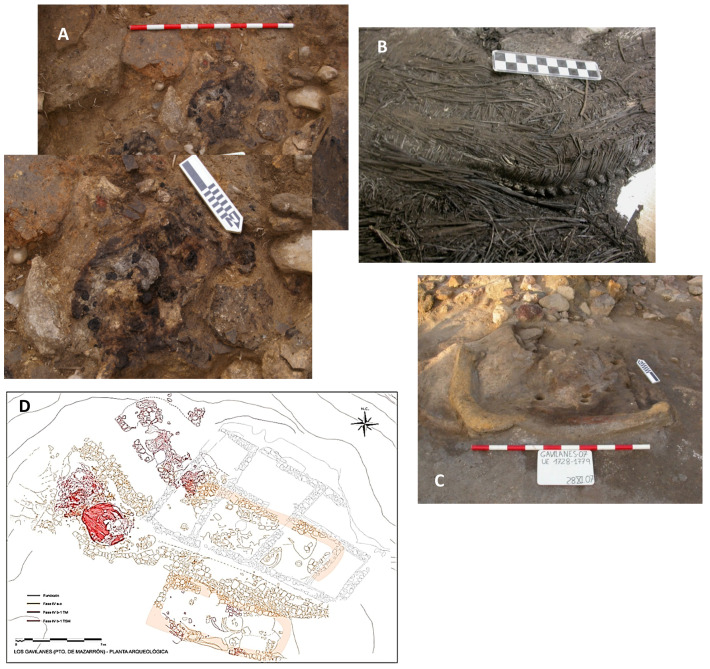
Roasting structure and esparto basket from Punta de los Gavilanes, Phase IV. (**A**) Roasting structure within a 1TM house; (**B**) Esparto basket within a 2TM building; (**C**) Domestic drying/smoking facility inside a 1TS house; (**D**) Urban layout of the 1TM and 1TS houses on the upper and middle terraces. *Image source: Gavilanes Project Archive*.

**Figure 4 genes-16-01477-f004:**
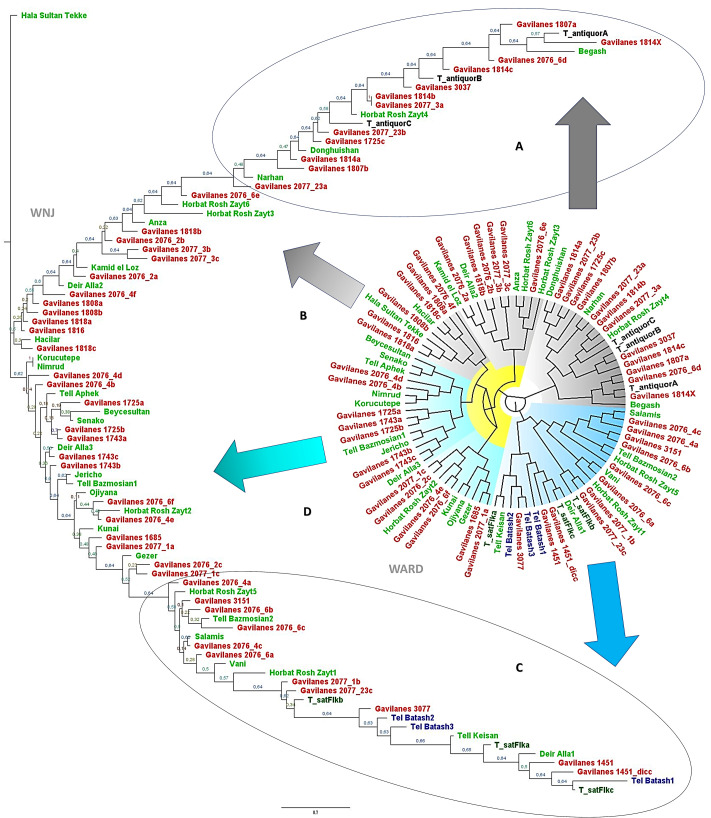
Multivariate morphometric comparison of archeological *Triticum* grain-specimens from Punta de los Gavilanes with reference taxa *T. turgidum* subsp. *parvicoccum* and *T. aestivum* subsp. *compactum* var*. antiquorum*, alongside comparative specimens from Neolithic and Bronze Age archeological contexts. The dendrograms illustrate taxonomic affinities based on comprehensive grain biometric parameters utilizing the following complementary clustering methodologies: Weighted Neighbor Joining (WNJ) and Ward’s minimum variance (WARD). Specimens are categorized into four morphological groupings as follows: (**A**) well-differentiated *T. aestivum* subsp. *compactum* var*. antiquorum*; (**B**) less morphologically distinct grain-specimens showing transitional characteristics; (**C**) well-differentiated *T. turgidum* subsp. *parvicoccum*; and (**D**) less differentiated specimens exhibiting intermediate morphological attributes. This analysis facilitates critical assessment of taxonomic boundaries within archaeobotanical assemblages and elucidates potential evolutionary relationships between these morphologically similar Bronze Age wheat taxa. Color codes: Red, 48 archeological samples from Punta de los Gavilanes; Light Green, other archeological caryopses (Table 1); Blue, type material of *T. turgidum* subsp. *parvicoccum* from Tell Batash [40,44]; Black, type material of *T. aestivum* subsp. *compactum* var*. antiquorum*, from Neolithic sites in Switzerland [25,26,49]; Deep Green, archeological caryopses morphologically similar to *T. turgidum* subsp. *parvicoccum* analyzed by Flaksberger [48].

**Figure 5 genes-16-01477-f005:**
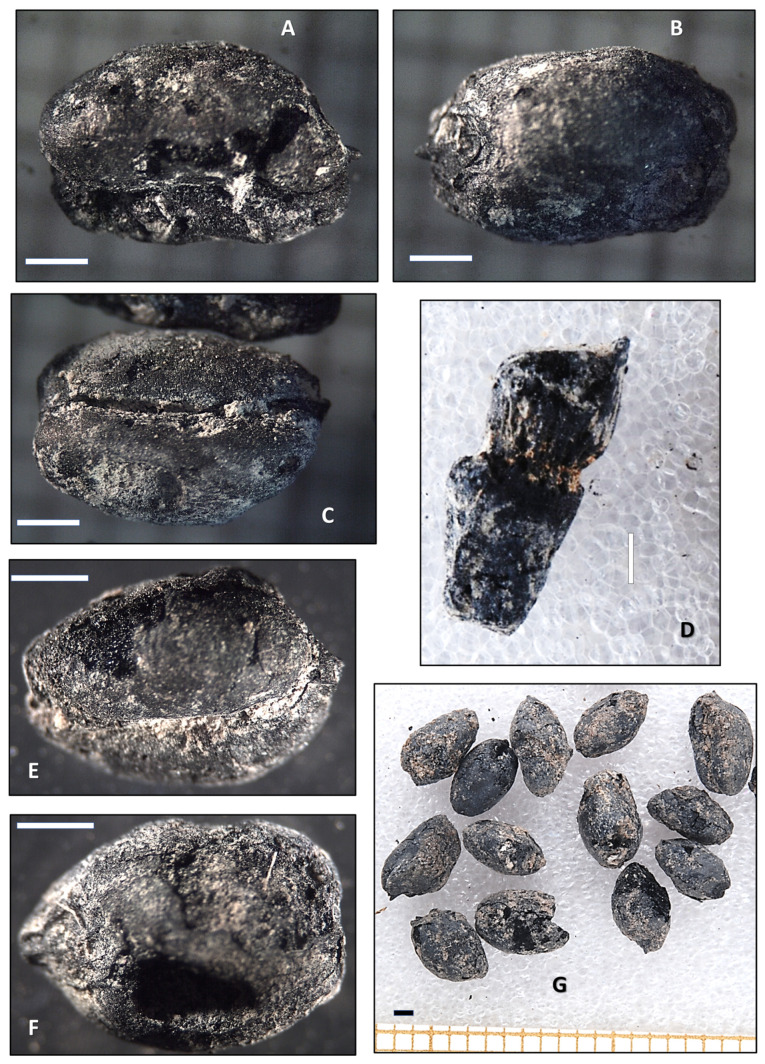
Charred carpological remains of small-grained wheat from different archeological samples. (**A**–**C**) Charred wheat grains from Sample 1814 (phase IVa1 TM), *T. sphaerococcum* Percival subsp. *sphaerococcum*, exhibiting morphological variability and surface alterations. (**D**) Fragment of a rachis internode and node from a tetraploid wheat (Sample 2074_23, phase Iva2 TM), a dense-eared form. Its morphology suggests it may belong to a free-threshing wheat (e.g., *T. turgidum* subsp*. durum*) or a hulled wheat (e.g., late-domesticated *T. turgidum* subsp*. dicoccum*). Further microscopic analysis would be required to confirm its classification as naked wheat. (**E**,**F**) Additional wheat grains from Sample 1807 (phase IVa2 TS), *T. sphaerococcum* Percival subsp. *sphaerococcum*, showing small and compact morphologies. (**G**) Cluster of wheat grains from Sample 2076_4 (phase IVa1 TM), *T. sphaerococcum* Percival subsp. *sphaerococcum,* highlighting size uniformity and preservation state. Scale bar = 1 mm. Images by D. Rivera.

**Figure 6 genes-16-01477-f006:**
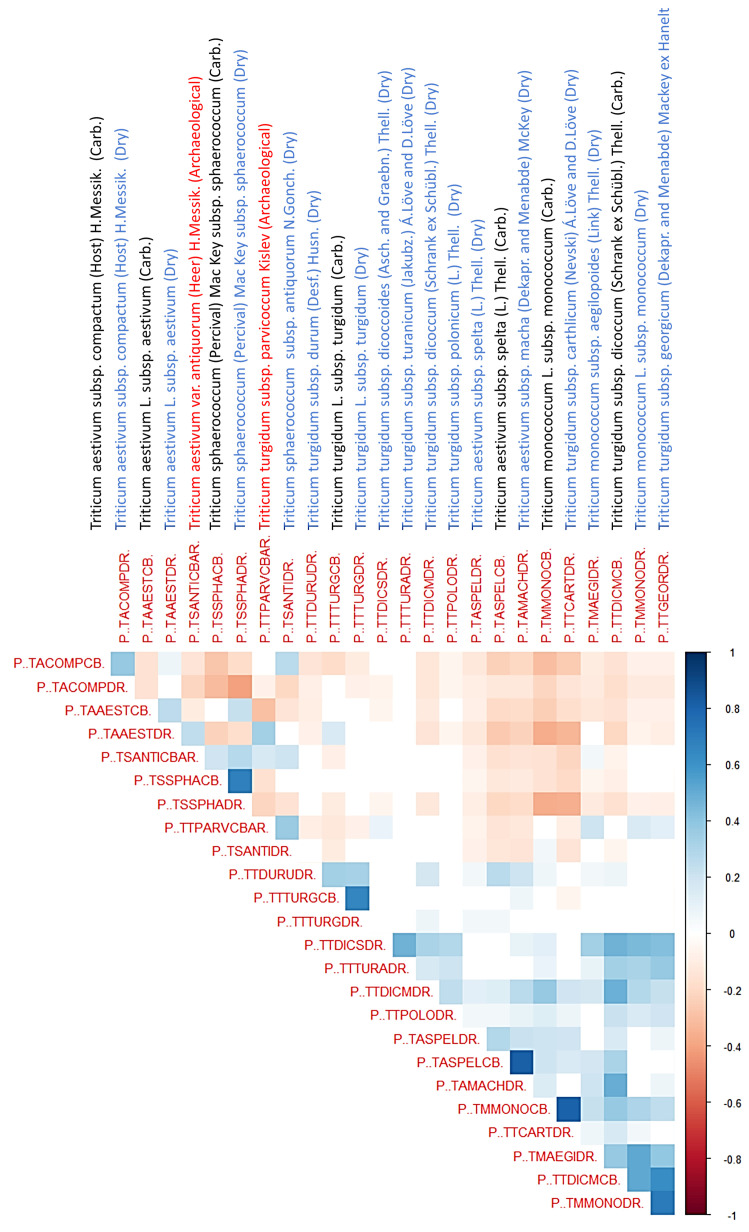
Correlation matrix of Random Forest classification probabilities for archeological wheat grain identification (n = 2463) revealing morphological relationships across modern taxa, archaeobotanical taxa, and preservation states. Blue vertically rotated labels indicate dry reference specimens, black denotes experimentally carbonized references, and red indicates archeological taxa. Strong positive correlations (dark blue, r > 0.6) between taxa pairs indicate morphological similarity leading to classification uncertainty, while strong negative correlations (dark red, r < −0.6) reveal distinctive morphological features enabling reliable taxonomic separation. The correlation patterns highlight both the challenges in distinguishing morphologically similar wheat taxa (particularly among *Triticum* subspecies) and the diagnostic potential of certain morphological features. This quantitative assessment provides a framework for evaluating classification confidence and identifying potential misidentification risks in archaeobotanical wheat assemblages.

**Figure 7 genes-16-01477-f007:**
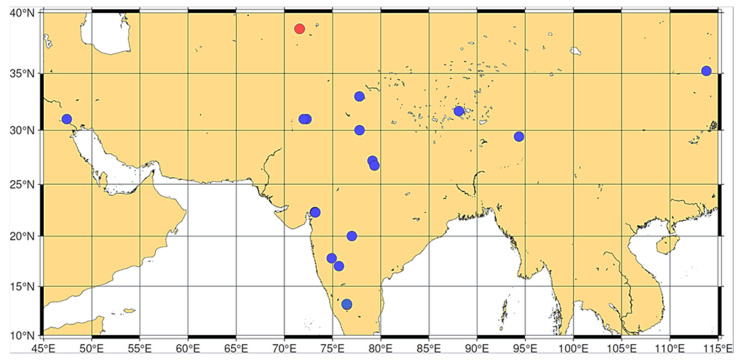
Distribution map for modern *Triticum sphaerococcum*. Red dot: *T. sphaerococcum* subsp. *antiquorum*. Blue dots: *T. sphaerococcum* subsp. *sphaerococcum.* Map: Francisco Alcaraz.

**Figure 8 genes-16-01477-f008:**
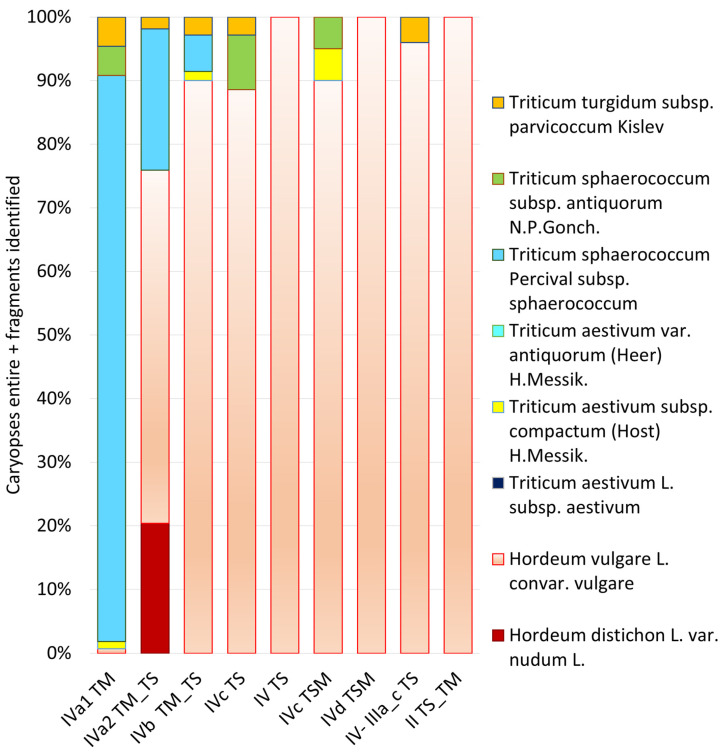
Diachronic distribution of cereal crops across archaeological phases at Punta de los Gavilanes (southeastern Iberia). The stacked bar chart illustrates the relative proportions of different wheat taxa (*Triticum* spp.) and barley varieties (*Hordeum* spp.) recovered from each chronological phase, arranged from earliest (IVa1 TM, 2290 BCE) to latest with radiocarbon data (IVc TSM, 1770 BCE) and later culturally dated (II TS_TM). Note the dramatic shift from wheat-dominated agriculture in the earliest phase to barley-dominated cultivation in later phases, with *Hordeum vulgare* becoming the predominant cereal by phase IVd TSM, linked likely to a gradual salinization process. This transformation suggests significant agricultural adaptation throughout the site’s occupation sequence.

**Figure 9 genes-16-01477-f009:**
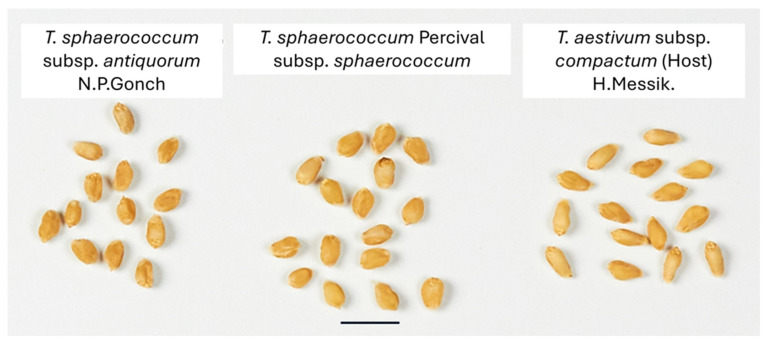
Comparison of modern globose and subglobose wheat caryopses of the two T. sphaerococcum subspecies, and those of club wheat (*T. aestivum* subsp. *compactum*). Scale bar: 10 mm. Image by Nikolay Goncharov.

**Table 2 genes-16-01477-t002:** Taxonomic identification of archaeobotanical wheat (*Triticum*) caryopses recovered from the Argaric archeological sites of Punta de Gavilanes (Murcia, Spain) and Almizaraque (Almería, Spain). Each seed is allocated to the taxon with highest probability among those with *p* ≥ 0.7, based on Random Forest and logistic regression (Logit) classification models. Results indicate *T. sphaerococcum* subsp. *sphaerococcum* as the predominant taxon with varying distributions of other wheat species, with a notable dominance of hexaploid.

Taxon	IP	NS Gavilanes RF		NS Gavilanes LO		NS Almizaraque RF		NS Almizaraque LO	
		(*p* > 0.7)	(*p* ≤ 0.7)	(*p* > 0.7)	(*p* ≤ 0.7)	(*p* > 0.7)	(*p* ≤ 0.7)	(*p* > 0.7)	(*p* ≤ 0.7)
**Desiccated identified with higher probability**									
*T. aestivum* L. subsp. *aestivum*	6x	0	1	0	4	16	28	0	113
*T. aestivum* subsp. *compactum* (Host) H.Messik.	6x	1	1	0	1	15	14	0	12
*T. aestivum* subsp. *macha* (Dekapr. and Menabde) McKey	6x	1	0	0	0	0	0	0	0
*T. sphaerococcum* Percival subsp. *sphaerococcum* (=*T. aestivum* subsp. *sphaerococcum* (Percival) Mac Key)	6x	18	5	22	3	151	13	130	23
*T. sphaerococcum* subsp. *antiquorum* N.P.Gonch.,	6x	8	1	1	2	39	17	14	15
*T. turgidum* subsp. *dicoccoides* (Asch. and Graebn.) Thell.	4x	0	0	1	0	0	0	6	0
*T. turgidum* subsp. *carthlicum* (Nevski) Á.Löve and D.Löve	4x	0	0	0	0	7	21	0	1
*T. turgidum* subsp. *turanicum* (Jakubz.) Á.Löve and D.Löve	4x	0	0	1	0	0	0	0	0
**Carbonized identified with higher probability**									
*T. aestivum* L. subsp. *aestivum*	6x	0	1	0	2	13	46	1	14
*T. aestivum* subsp. *compactum* (Host) H.Messik.	6x	0	0	3	2	24	57	14	20
*T. sphaerococcum* Percival subsp. *sphaerococcum* (=*T. aestivum* subsp. *sphaerococcum* (Percival) Mac Key)	6x	1	0	0	0	3	3	9	1
*T. turgidum* L. subsp*. turgidum*	4x	0	1	0	0	0	0	0	0
*T. monococcum* L. subsp. *monococcum*	2x	0	0	0	0	0	2	10	0
**Archaeobotanical identified with higher probability**									
*T. turgidum* subsp. *parvicoccum* Kislev	4x	1	3	6	0	5	35	70	57
*T. aestivum* var. *antiquorum* (Heer) H.Messik.,	6x	1	2	0	0	0	8	0	0

Abbreviations: IP, inferred ploidy based on the taxonomical allocation of the seed; NS, number of seeds analyzed.

**Table 3 genes-16-01477-t003:** Comparative analysis of machine learning techniques for the tentative identification of specific archeological grains of small-grained, free-threshing domesticated wheats from Punta de los Gavilanes (Murcia, Spain). Results are presented from Random Forest (RF) and logistic regression (LO) including the most probable identity among the taxa and statuses incorporated in the training set together with the estimated probability assigned by the model.

Sample	Date Calibrated Year	Taxa and Status of the Reference	RF	Taxa and Status of the Reference	LO
Gavilanes 1451	1890 BC (93.8%)	*T. aestivum* subsp. *compactum* (Host) H.Messik. (Dry)	0.56	*T. aestivum* subsp. *compactum* (Host) H.Messik. (Dry)	0.32
Gavilanes 1451_dicc		*T. aestivum* subsp. *macha* (Dekapr. and Menabde) McKey (Dry)	0.75	*T. turgidum* subsp*. turanicum* (Jakubz.) Á.Löve and D.Löve	1.00
Gavilanes 1685		*T. sphaerococcum* subsp. *antiquorum* N.P.Gonch. (Dry)	0.64	*T. turgidum* subsp. *parvicoccum* Kislev (Archaeo)	0.96
Gavilanes 1725a		*T. sphaerococcum* subsp. *antiquorum* N.P.Gonch. (Dry)	0.91	*T. turgidum* subsp. *parvicoccum* Kislev (Archaeo)	1.00
Gavilanes 1725b		*T. sphaerococcum* subsp. *antiquorum* N.P.Gonch. (Dry)	0.88	*T. turgidum* subsp. *parvicoccum* Kislev (Archaeo)	0.96
Gavilanes 1725c		*T. sphaerococcum* Percival subsp*. sphaerococcum* (Dry)	0.70	*T. sphaerococcum* Percival subsp*. sphaerococcum* (Dry)	0.98
Gavilanes 1743a		*T. sphaerococcum* subsp*. antiquorum* N.P.Gonch. (Dry)	0.97	*T. turgidum* subsp. *parvicoccum* Kislev (Archaeo)	0.96
Gavilanes 1743b		*T. sphaerococcum* subsp*. antiquorum* N.P.Gonch. (Dry)	0.80	*T. turgidum* subsp. *parvicoccum* Kislev (Archaeo)	0.73
Gavilanes 1743c		*T. sphaerococcum* subsp*. antiquorum* N.P.Gonch. (Dry)	0.90	*T. sphaerococcum* Percival subsp*. sphaerococcum* (Dry)	0.57
Gavilanes 1807a		*T. sphaerococcum* Percival subsp*. sphaerococcum* (Dry)	1.00	*T. sphaerococcum* Percival subsp*. sphaerococcum* (Dry)	0.99
Gavilanes 1807b		*T. sphaerococcum* Percival subsp*. sphaerococcum* (Dry)	1.00	*T. sphaerococcum* Percival subsp*. sphaerococcum* (Dry)	0.99
Gavilanes 1808a	2035 BC (68.2%)	*T. sphaerococcum* Percival subsp*. sphaerococcum* (Dry)	0.54	*T. sphaerococcum* Percival subsp*. sphaerococcum* (Dry)	0.81
Gavilanes 1808b	2035 BC (68.2%)	*T. sphaerococcum* Percival subsp*. sphaerococcum* (Dry)	0.97	*T. sphaerococcum* Percival subsp*. sphaerococcum* (Dry)	0.87
Gavilanes 1814a		*T. aestivum* var. *antiquorum* (Heer) H.Messik. (Archaeo)	0.74	*T. sphaerococcum* Percival subsp*. sphaerococcum* (Dry)	0.92
Gavilanes 1814b		*T. sphaerococcum* Percival subsp*. sphaerococcum* (Dry)	1.00	*T. sphaerococcum* Percival subsp*. sphaerococcum* (Dry)	0.99
Gavilanes 1814c		*T. sphaerococcum* Percival subsp*. sphaerococcum* (Dry)	0.98	*T. sphaerococcum* Percival subsp*. sphaerococcum* (Dry)	0.99
Gavilanes 1814X		*T. sphaerococcum* Percival subsp. *sphaerococcum* (Carbonized)	0.91	*T. turgidum* subsp. *dicoccoides* (Asch. and Graebn.) Thell. (Dry)	1.00
Gavilanes 1816		*T. sphaerococcum* Percival subsp*. sphaerococcum* (Dry)	0.63	*T. sphaerococcum* Percival subsp*. sphaerococcum* (Dry)	0.75
Gavilanes 1818a	2140 BC (95.4%)	*T. sphaerococcum* Percival subsp*. sphaerococcum* (Dry)	0.91	*T. sphaerococcum* Percival subsp*. sphaerococcum* (Dry)	0.87
Gavilanes 1818b	2140 BC (95.4%)	*T. sphaerococcum* Percival subsp*. sphaerococcum* (Dry)	0.94	*T. sphaerococcum* Percival subsp*. sphaerococcum* (Dry)	0.95
Gavilanes 1818c	2140 BC (95.4%)	*T. turgidum* subsp. *parvicoccum* Kislev (Archaeo)	0.68	*T. aestivum* subsp. *compactum* (Host) H.Messik. (Carbonized)	0.66
Gavilanes 2076_2a	2290 BC (82.8%)	*T. sphaerococcum* Percival subsp*. sphaerococcum* (Dry)	0.65	*T. sphaerococcum* Percival subsp*. sphaerococcum* (Dry)	0.88
Gavilanes 2076_2b	2290 BC (82.8%)	*T. sphaerococcum* Percival subsp*. sphaerococcum* (Dry)	0.96	*T. sphaerococcum* Percival subsp*. sphaerococcum* (Dry)	0.89
Gavilanes 2076_2c	2290 BC (82.8%)	*T. aestivum* L. subsp. *aestivum* (Carbonized)	0.71	*T. aestivum* L. subsp. *aestivum* (Dry)	0.35
Gavilanes 2076_4a		*T. turgidum* subsp. *parvicoccum* Kislev (Archaeo)	0.60	*T. aestivum* subsp. *compactum* (Host) H.Messik. (Carbonized)	0.85
Gavilanes 2076_4b		*T. turgidum* subsp. *parvicoccum* Kislev (Archaeo)	0.59	*T. aestivum* subsp. *compactum* (Host) H.Messik. (Carbonized)	0.49
Gavilanes 2076_4c		*T. aestivum* L. subsp. *aestivum* (Dry)	0.48	*T. aestivum* subsp. *compactum* (Host) H.Messik. (Carbonized)	0.73
Gavilanes 2076_4d		*T. sphaerococcum* Percival subsp. *sphaerococcum* (Dry)	0.79	*T. aestivum* L. subsp. *aestivum* (Carbonized)	0.45
Gavilanes 2076_4e		*T. aestivum* L. subsp. *aestivum* (Carbonized)	0.77	*T. aestivum* L. subsp. *aestivum* (Carbonized)	0.40
Gavilanes 2076_4f		*T. sphaerococcum* Percival subsp*. sphaerococcum* (Dry)	0.82	*T. sphaerococcum* Percival subsp*. sphaerococcum* (Dry)	0.90
Gavilanes 2076_6a		*T. turgidum* subsp. *parvicoccum* Kislev (Archaeo)	0.76	*T. aestivum* subsp. *compactum* (Host) H.Messik. (Carbonized)	0.91
Gavilanes 2076_6b		*T. sphaerococcum* subsp. *antiquorum* N.P.Gonch. (Dry)	0.93	*T. sphaerococcum* subsp. *antiquorum* N.P.Gonch. (Dry)	0.59
Gavilanes 2076_6c		*T. turgidum* L. subsp. *turgidum* (Carbonized)	0.39	*T. aestivum* L. subsp. *aestivum* (Dry)	0.27
Gavilanes 2076_6d		*T. sphaerococcum* Percival subsp*. sphaerococcum* (Dry)	0.91	*T. sphaerococcum* Percival subsp*. sphaerococcum* (Dry)	1.00
Gavilanes 2076_6e		*T. sphaerococcum* Percival subsp*. sphaerococcum* (Dry)	1.00	*T. sphaerococcum* Percival subsp*. sphaerococcum* (Dry)	0.99
Gavilanes 2076_6f		*T. sphaerococcum* Percival subsp*. sphaerococcum* (Dry)	0.86	*T. sphaerococcum* Percival subsp*. sphaerococcum* (Dry)	0.34
Gavilanes 2077_1a		*T. sphaerococcum* subsp. *antiquorum* N.P.Gonch. (Dry)	0.80	*T. turgidum* subsp. *parvicoccum* Kislev (Archaeo)	0.75
Gavilanes 2077_1b		*T. aestivum* var. *antiquorum* (Heer) H.Messik. (Archaeo)	0.36	*T. aestivum* L. subsp. *aestivum* (Dry)	0.30
Gavilanes 2077_1c		*T. sphaerococcum* Percival subsp*. sphaerococcum* (Dry)	0.43	*T. sphaerococcum* Percival subsp*. sphaerococcum* (Dry)	0.25
Gavilanes 2077_23a	2140 BC (68.2%)	*T. aestivum* var*. antiquorum* (Heer) H.Messik. (Archaeo)	0.69	*T. sphaerococcum* Percival subsp*. sphaerococcum* (Dry)	0.98
Gavilanes 2077_23b	2140 BC (68.2%)	*T. sphaerococcum* Percival subsp*. sphaerococcum* (Dry)	0.96	*T. sphaerococcum* Percival subsp*. sphaerococcum* (Dry)	1.00
Gavilanes 2077_23c	2140 BC (68.2%)	*T. aestivum* L. subsp. *aestivum* (Carbonized)	0.50	*T. sphaerococcum* subsp. *antiquorum* N.P.Gonch. (Dry)	0.30
Gavilanes 2077_3a		*T. sphaerococcum* Percival subsp*. sphaerococcum* (Dry)	1.00	*T. sphaerococcum* Percival subsp*. sphaerococcum* (Dry)	0.99
Gavilanes 2077_3b		*T. sphaerococcum* Percival subsp*. sphaerococcum* (Dry)	1.00	*T. sphaerococcum* Percival subsp*. sphaerococcum* (Dry)	0.95
Gavilanes 2077_3c		*T. sphaerococcum* Percival subsp*. sphaerococcum* (Dry)	0.74	*T. sphaerococcum* Percival subsp*. sphaerococcum* (Dry)	0.99
Gavilanes 3037		*T. sphaerococcum* Percival subsp*. sphaerococcum* (Dry)	0.91	*T. sphaerococcum* Percival subsp*. sphaerococcum* (Dry)	1.00
Gavilanes 3077		*T. aestivum* subsp. *compactum* (Host) H.Messik. (Dry)	0.75	*T. aestivum* L. subsp. *aestivum* (Dry)	0.23
Gavilanes 3151		*T. sphaerococcum* subsp*. antiquorum* N.P.Gonch. (Dry)	1.00	*T. sphaerococcum* subsp. *antiquorum* N.P.Gonch. (Dry)	0.85

Note: cf. Appendix A. Reference dimensions for *T. aestivum* var. *antiquorum* (Heer) H.Messik. are derived from Flaksberger [48], specifically from Neolithic dwellings of Swiss lakes. Reference dimensions for *T. turgidum* subsp. *parvicoccum* Kislev are derived from Kislev [43], specifically from Bronze Age levels at Tell Batash, Israel. Those identification with lower support, below 0.7, are underlined.

**Table 4 genes-16-01477-t004:** Comparative analysis of classical and machine learning techniques for the tentative identification of archeological samples of small-grained, free-threshing domesticated wheats from Almizaraque, c. 2000 BC. (Almería, Spain).

Samples	Prior Identification by Tellez and Ciferri	According to Random Forest	According to Logistic Regression
Caja 1859. Muestra 10.	*T. aestivum* subsp. *compactum* (Host) H.Messik	*T. sphaerococcum* Percival subsp. *sphaerococcum* (Dry, mean 0.84, sd 0.26)	*T. sphaerococcum* Percival subsp. *sphaerococcum* (Dry, mean 0.8, sd 0.28)
Caja 1859. Muestra 11.	*T. aestivum* L. subsp. *aestivum*	*T. turgidum* subsp. *parvicoccum* Kislev (Archaeo, mean 0.45, sd 0.12)	*T. turgidum* subsp. *parvicoccum* Kislev (Archaeo, mean 0.82, sd 0.27)
Caja 1859. Muestra 9.	*T. aestivum* L. subsp. *aestivum*	*T. aestivum* L. subsp. *aestivum* (Carbonized, mean 0.39, sd 0.12)	*T. aestivum* L. subsp. *aestivum* (Dry, mean 0.36, sd 0.09)
Casa 32 Caja 1857. Muestra 4.	*T. aestivum* subsp. *compactum* (Host) H.Messik	*T. sphaerococcum* Percival subsp. *sphaerococcum* (Dry, mean 0.88, sd 0.22)	*T. sphaerococcum* Percival subsp. *sphaerococcum* (Dry, mean 0.82, sd 0.30)
Casa 32 Caja 1857. Muestra 5.	*T. aestivum* subsp. *compactum* (Host) H.Messik	*T. aestivum* L. subsp. *aestivum* (Carbonized, mean 0.33, sd 0.2)	*T. aestivum* L. subsp. *aestivum* (Dry, mean 0.38, sd 0.08)
Casa 32 Caja 1857. Muestra 6.	*T. aestivum* L. subsp. *aestivum*	*T. turgidum* subsp. *parvicoccum* Kislev (Archaeo, mean 0.52, sd 0.13)	*T. turgidum* subsp. *parvicoccum* Kislev (Archaeo, mean 0.86, sd 0.21)
Casa 40 Caja 1854. Muestra 1.	*T. aestivum* L. subsp. *aestivum*	*T. aestivum* subsp. *compactum* (Host) H.Messik. (Carbonized, mean 0.42, sd 0.26)	*T. aestivum* L. subsp. *aestivum* (Dry, mean 0.36, sd 0.09)
Casa 40 Caja 1854. Muestra 2.	*T. aestivum* subsp. *compactum* (Host) H.Messik	*T. sphaerococcum* Percival subsp. *sphaerococcum* (Dry, mean 0.89, sd 0.25)	*T. sphaerococcum* Percival subsp. *sphaerococcum* (Dry, mean 0.92, sd 0.22)
Casa 41 Caja 1854. Vasija 8 Muestra 3.	*T. aestivum* subsp. *compactum* (Host) H.Messik	*T. sphaerococcum* subsp. *antiquorum* N.P.Gonch. (Dry, mean 0.47, sd 0.31)	*T. turgidum* subsp. *parvicoccum* Kislev (Archaeo, mean 0.54, sd 0.2)
Casa 41 Caja 1857. Muestra 7.	*T. aestivum* L. subsp. *aestivum*	*T. aestivum* subsp. *compactum* (Host) H.Messik. (Carbonized, mean 0.33, sd 0.23)	*T. aestivum* L. subsp. *aestivum* (Dry, mean 0.36, sd 0.1)
Casa 41 Caja 1857. Muestra 8.	*T. aestivum* subsp. *compactum* (Host) H.Messik	*T. sphaerococcum* Percival subsp. *sphaerococcum* (Dry, mean 0.92, sd 0.16)	*T. sphaerococcum* Percival subsp. *sphaerococcum* (Dry, mean 0.89, sd 0.19)

Note: Tentative identification by Tellez and Ciferri [68] is based on dimensional comparisons of 50 grains per sample based on the following key data: general shape and contour of the grain, morphology near the germ and apex, absolute and relative grain dimensions, presence of “brush” at the apex, shape of the dorsal surface, crease and lateral lobes, form of the germ, and crease perforation. They are compared here with a tentative identification based on grain dimensions and allometric proportions with probabilities calculated for each individual grain using Random Forest and logistic regression.

## Data Availability

The Excel book with crude measurements and RF and LOG calculated probabilities is available at GitHub (https://github.com/drivera2001/Triticum-Gavilanes-T.-parvicoccum-and-T.-antiquorum-training-and-results: Basic training sets and results of identification, using Logit and random Forest, of archaeobotanical wheat caryopses).

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
