# Peer review of "Revealing Ancient Wheat Phylogenetic Diversity: Machine Learning and Logistic Regression Identify Triticum sphaerococcum in Bronze Age Iberia"

_genes, 2025, doi:10.3390/genes16121477_

Round 1
Reviewer 1 Report
Comments and Suggestions for Authors
Dear sir,
I have some comments:
- Line 686-694. T. sphaerococcum could have reached Spain from its putative center of origin, the Fertile Crescent, and not from Central Asia and India, where they stay longer, and until now, but it is farther. Actually, there is no reference to where T. sphaer... originated.
- Line 706-711. Possibly preference for other wheat species, with more yield, better quality for bread or other food, etc.
- Author contributions: I think you must write only the initials of names and family names.
- References: Zohary´s book, 'domestication of plants in the old World' is repeated twice. Please remove one.
- Table S1. In the column Sample, the first letter must be in italics.
Best regards
Reviewer 2 Report
Comments and Suggestions for Authors
The manuscript entitled “Revealing Ancient Wheat Phylogenetic Diversity: Machine Learning and Logistic Regression Identify Triticum sphaerococcum in Bronze Age Iberia” presents a novel integration of computational morphometrics and archaeobotanical data to trace ancient wheat diversity. Using Random Forest and logistic regression models on morphometric datasets from both modern and archaeological wheat grains, the authors identify Triticum sphaerococcum—today restricted to Central and South Asia—in Bronze Age Iberian contexts. The study’s main strengths are its innovative methodological framework, robust sample size, and the interdisciplinary collaboration bridging archaeobotany, plant taxonomy, and computational modeling. The work makes a valuable contribution to understanding crop phylogeography and genetic erosion in cultivated plants. I have some comments:
1. How were the Random Forest and logistic regression models validated beyond internal dataset performance?
2. The manuscript provides strong regional insight but could better contextualize findings within broader Eurasian archaeobotanical data. How do these results compare to other Western Mediterranean sites?
3. The authors might consider discussing the implications of these findings for current wheat conservation and breeding programs.
4. The authors should add a concise paragraph in the Discussion explicitly titled “Limitations”.
Reviewer 3 Report
Comments and Suggestions for Authors
- The figures/tables should be inserted into the main text close to their first citation. When inserting the figure/table in the text, first mention the figure/table in the text, then the concrete figure/table and then the description of the results presented on this figure/table.
- The figures/tables in the section Discussion could be removed in the section Results.
- The Section conclusion needs to be more brief. Lines 1081 to 1098 could be removed at the end of the section Discussion.
- The part of the short explanatory caption/legend of Table 1 (Note, Lines 274-278) could be removed in the text. Same suggestion for Figure 4 (Lines 403-416, “The dendrograms…” could be removed in the text.). Same suggestion for Figures 5 and 6.
- In Line 1061, "Iberian Peninsula" should be with a capital letter.
- The quality of the manuscript would be improved if the Discussion included the interpretation of a comprehensive analysis of machine learning and logistic regression methodologies (this interpretation is included in the Section Results).
